# MEASURING LLM NOVELTY AS THE FRONTIER OF ORIGINAL AND HIGH-QUALITY OUTPUT

**Vishakh Padmakumar**
New York University
vishakh@nyu.edu

**Chen Yueh-Han**
New York University
yueh.han.chen@nyu.edu

**Jane Pan**
New York University
jane.pan@nyu.edu

**Valerie Chen**
Carnegie Mellon University
vchen2@andrew.cmu.edu

**He He**
New York University
hhe@nyu.edu

## ABSTRACT

As large language models (LLMs) are increasingly used for ideation and scientific discovery, it is important to evaluate their ability to generate novel output. Prior work evaluates novelty as originality with respect to model training data, but original outputs may be of low quality. In contrast, non-expert judges more reliably score quality but may favor memorized outputs, limiting the reliability of human preference as a metric. We introduce a new novelty metric for LLM generations that balances originality and quality—the harmonic mean of the fraction of $n$-grams unseen during training and a task-specific quality score. Using this framework, we identify trends that affect the novelty of generations from three families of open-data models (OLMo, OLMo-2, and Pythia) on three creative tasks: story completion, poetry writing, and creative tool use. We find that model-generated text from some base LLMs is less novel than human-written text from the internet. However, increasing model scale and post-training reliably improves novelty due to improvements in output quality. We also find that improving the base model at the same scale (e.g., OLMo 7B to OLMo-2 7B) leads to higher novelty due to higher originality. Finally, we observe that inference-time methods, such as prompting and providing novel in-context examples, have a much smaller effect on novelty, often increasing originality at the expense of quality. This highlights the need for further research into more effective elicitation strategies as we use models for creative applications.

## 1 INTRODUCTION

As large language models (LLMs) are increasingly used for creative tasks (Wan et al., 2024; Haase & Pokutta, 2024; Moruzzi & Margarido, 2024) and scientific discovery (Gottweis et al., 2025; Feng et al., 2024), it is important to evaluate their ability to generate novel output. Past work measures novelty by memorization; that is, whether text fragments appear in training data (McCoy et al., 2023; Merrill et al., 2024; Lu et al., 2024a). However, originality alone is not sufficient. Consider a scenario in which a user asks for suggestions from an LLM when writing a poem (Figure 1). The output may be highly original, but of poor quality. To identify high-quality outputs, leaderboards like Chatbot Arena (Chiang et al., 2024) collect and aggregate human preference judgments. However, these are unsatisfactory measures of novelty as a novice judge might score output highly, not knowing that it is copied verbatim from the pre-training data.

Ideally, models should generate output that uses expressive and figurative language without reproducing the training data. In this paper, we argue that these two facets must be jointly considered. We propose to measure novelty as the harmonic mean of *originality* (measured by the fraction of unseen $n$-grams in a generation) and *quality* according to task-specific measures (Section 2.1). We use this metric to answer the following research questions.

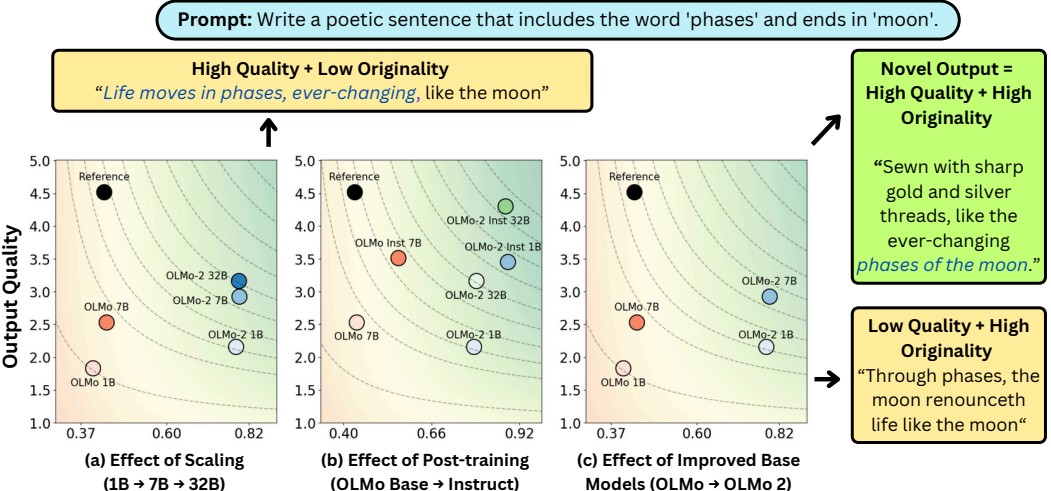

**Originality = Unseen N-gram Fraction**

Figure 1: We evaluate LLMs' ability to generate novel text, defined as high-quality responses that avoid reproducing higher-order $n$-grams from training data (highlighted in *blue*). Novelty is measured as the harmonic mean of unseen $n$-gram fraction ($x$-axis) and output quality ($y$-axis) (Section 2.1). Contour lines denote equal novelty in each plot. We find that: (a) scaling models and (b) post-training increase novelty through improved quality, while (c) stronger base models (e.g., OLMo 1 to OLMo 2) improve novelty by generating more original output (Section 3). Inference-time methods (e.g., novel ICL examples, Denial Prompting) have limited effect on shifting the novelty frontier (Section 4).

**What factors affect the novelty of LLM output?** We analyze generations from three families of open-data models—OLMo (Groeneveld et al., 2024), OLMo-2 (OLMo et al., 2024), and Pythia (Biderman et al., 2023b)—to identify factors that affect LLM novelty across three creativity-focused tasks (Section 2.2), ranging from story completion (Eldan & Li, 2023) to poetry writing (Chakrabarty et al., 2022) to creative tool use (Tian et al., 2024). We find that scaling LLMs results in more novel output (OLMo 1B to 7B), though the gains plateau at higher scales (OLMo-2 7B to 32B). Here, the improvement comes from higher quality output while originality remains stable within a model family. Post-training also consistently leads to higher novelty than base models due to higher quality and similar originality across all model scales. Finally, improving the underlying base model at the same scale (e.g., OLMo to OLMo 2) increases novelty by improving originality (Section 3).

**Can we elicit more novel outputs from LLMs at inference time?** We investigate whether inference-time methods (e.g., changing the decoding strategy or prompt) elicit more novel output. We find that while increasing the sampling temperature initially boosts novelty by increasing originality, these gains can be quickly outweighed by a decline in quality (Section 4.1). For prompting base LLMs (Section 4.2), we use high-novelty in-context examples; and for post-trained models (Section 4.3), we experiment with *asking for novelty* and *denial prompting* (Lu et al., 2024b). These methods have a smaller effect on novelty by generating slightly more original output while paying a cost in quality.

Our main contribution is a metric for studying novelty that allows comparison of models from different families, scales, and training methods on an equal footing, which helps uncover the factors that affect novel output generation. Using this measure, we identify trade-offs between originality and quality, emphasizing the importance of considering both together. We find that scaling, alignment, and improving the underlying base LLM can push the Pareto frontier of novelty, whereas inference-time methods yield only limited gains, motivating further research into more effective elicitation strategies. While we focus on open-data models, which allow us to accurately evaluate originality, our analysis can also be extended naturally to black-box models, where providers can directly report aggregated novelty scores without exposing proprietary data (Section 6). This approach helps the community track novelty over time, place advances in creative and scientific context, and evaluate true generalization for AI safety. We release the dataset of over 5000 LLM generations, with quality scores and copied n-grams to facilitate research along this direction.[1]

---

[1] We will make our code and model outputs available upon publication.

## 2 MEASURING NOVELTY OF LLM GENERATIONS

In this section, we present our evaluation method to measure the novelty of LLM generations. We introduce a new metric definition (Section 2.1), that we apply to a suite of creative task datasets (Section 2.2) and finally provide details about how we operationalize our definition (Section 2.3) for subsequent experiments (Section 3 and Section 4).

### 2.1 METRIC DEFINITION

We propose a measure of novelty that captures both originality (i.e., whether the text is different from the training data) and quality, ensuring that novel generations remain coherent and helpful to users.

**Novel output should be *original*.** We must first distinguish between content that is genuinely new, rather than reproducing the training data of the model. The de facto approach to measuring the originality of output is to calculate the fraction of higher-order $n$-grams which do not appear in pre- and post-training data of LLMs (McCoy et al., 2023; Elazar et al., 2024; Merrill et al., 2024; Lu et al., 2024a). This value can be seen as a distance metric with outputs containing more unseen $n$-grams as farther away from the training data and therefore more original. Following McCoy et al. (2023); Elazar et al. (2024); Merrill et al. (2024), we calculate $n$-gram *originality* as the proportion of $n$-grams in a generation that do not appear in a corpus $C$, where $C$ corresponds to the pre- and post-training corpora of the LLM used for generation. The tools used are detailed in Section 2.3.1.

**Novel outputs should be *high quality*.** Identifying original outputs alone is insufficient, since long-tail generations that are original may also be nonsensical. As such, we also desire outputs to be high-quality with respect to the user prompt. While we would ideally measure output quality using human annotations, large-scale human evaluation is impractical for benchmarking various ablations of model performance. Instead, we use LLM-as-a-judge evaluation to rate output quality, providing a scalable approximation of user preferences. Since measures of quality are highly task-specific, we provide the prompts used for each task in Section 2.2 and provide details about how we validate the reliability of automatic scoring in Section 2.3.2.

**Our novelty metric.** As illustrated by the example in Figure 1, existing metrics capture just a single dimension of novelty. For instance, metrics like *Creativity Index* (Lu et al., 2024a) and $n$-novelty (Merrill et al., 2024) only score originality and would incorrectly rank rare but poor-quality output highly. Meanwhile, benchmarks of output quality, like ChatBot Arena (Chiang et al., 2024) rely on human ratings, which might favourably judge the unoriginal answer.[2] To aggregate both dimensions into a single measure of novelty, we report the harmonic mean of quality (renormalized to a value between 0 and 1) and originality (as measured by the unseen $n$-gram fraction) of each generation, which correctly identifies truly novel generations.[3] We report average novelty on three tasks (Section 2.2), allowing us to compare different models and ablations of generation methods.[4]

### 2.2 CREATIVE TASKS

We evaluate the novelty of generations on three tasks: story completion, poetry writing, and creative tool use. We select these tasks because they are open-ended, with a wide range of valid responses that allow for varying novelty. Table 5 provides examples of each task.

**Story completion.** We use the TinyStories dataset (Eldan & Li, 2023) to evaluate model generated story endings. Following Yang et al. (2022), the model is provided with a prompt consisting of the first line of a story, which introduces the setting and characters, and must then complete the story. To score generation quality, we use an evaluation prompt that assigns points for correctly reusing and developing the introduced characters and plot elements, maintaining coherence, ensuring logical progression, and preserving grammatical correctness (Appendix E.1.1).

**Poetry writing.** We use the CoPoet dataset (Chakrabarty et al., 2022), where the model generates a single line of poetry in response to a given instruction about the content and literary devices to be

---

[2] We provide more examples in Table 3.

[3] We select the harmonic mean to penalize either originality or quality being low.

[4] In Section 6 we detail how our evaluation method can be used for black-box models as well as updated as the research community makes progress in measuring more high-quality measures of originality.

included. To score quality, we use an evaluation prompt that assigns points based on adherence to the instruction, correct use of specified literary devices, coherence, and grammaticality (Appendix E.1.2).

**Creative tool use.** We use the MacGyver dataset (Tian et al., 2024) of reasoning problems that require creative use of items to complete physical objectives. The model is prompted with the scenario and must generate a solution through innovative but feasible use of common objects. We score quality with a prompt that checks whether the proposed solution correctly utilizes the provided tools in a valid manner, and successfully resolves the given problem (Appendix E.1.3).

## 2.3 OPERATIONALIZING OUR NOVELTY METRIC

### 2.3.1 CALCULATING OUTPUT ORIGINALITY

We measure originality as the fraction of $n$-grams that do not appear in model training data. We calculate this using the WIMBD API (Elazar et al., 2024) and Infinigram (Liu et al., 2024; 2025), which index the pre- and post-training corpora of various open-data model families. Our experiments (Section 3 and Section 4) use the Pythia, OLMo, and OLMo-2 models which are covered by the indexes for the Pile (Gao et al., 2020), Dolma (Soldaini et al., 2024), Dolmino (OLMo et al., 2024), OLMo-Tulu SFT mixture (Ivison et al., 2023), OLMo-2-Preference mixture (OLMo et al., 2024), Tulu RLVR mixture (Lambert et al., 2024), and Ultrafeedback (Cui et al., 2024). This allows us to check whether the constituent $n$-grams of generations from the models appear in their training data. Following Merrill et al. (2024), we consider $n = 4$, 5, and 6, since smaller values result in nearly zero unseen $n$-grams, while larger values lead to almost all $n$-grams being unseen.

### 2.3.2 LLM-AS-A-JUDGE AS A MEASURE OF OUTPUT QUALITY

We use LLM-as-a-judge to approximate the measure of output quality from human annotators in a scalable manner. To ensure that we obtain reliable ratings, we perform a human study. We obtain three human annotations each for 100 examples for all three tasks from Upwork. The scoring rubric provided to annotators was the same as the 'prompt' used with the LLM (Appendix E). We find that inter-annotator agreement, measured by Krippendorff's alpha (Krippendorff, 2018), was 0.68 for CoPoet, 0.64 for MacGyver, and 0.59 for TinyStories, consistent with agreement levels reported for creative tasks in contemporary works (Li et al., 2025; Sawicki et al., 2025; Chiang & Lee, 2023; Chakrabarty et al., 2024). We then compare different LLMs and prompting setups (e.g., in-context examples, average of multiple runs) using the Spearman correlation of model-assigned quality scores to the average annotator ratings (Table 4 in Appendix D). We find that the highest average correlation—0.50 for CoPoet, 0.52 for TinyStories and 0.52 for MacGyver—is `o3-mini`, averaging the scores over 5 runs. For the rest of this paper, all scores of output quality use this setup.[5]

## 3 WHAT FACTORS AFFECT THE NOVELTY OF LLM OUTPUT?

### 3.1 EXPERIMENTAL SETUP

**Models.** We evaluate generations from three families of open-data models—OLMo (Groeneveld et al., 2024), OLMo-2 (OLMo et al., 2024) and Pythia (Biderman et al., 2023b). We evaluate the following models: (1) OLMo-1B and 7B, (2) OLMo-2-1B, 7B, 13B and 32B, (3) Pythia-6.9B and 12B[6], (4) Pythia-Deduped-1B, 2.8B, 6.9B and 12B (Pythia DDP)[7] Since these are base LLMs only pre-trained on the next-token objective, we provide 5 in-context learning (ICL) examples, randomly sampled from the validation split, to illustrate each task.[8] We also evaluate OLMo-7B-Instruct and OLMo-2-Instruct 1B, 7B, 13B, 32B to ablate the impact of post-training (with SFT+DPO for OLMo and SFT+DPO+RLVR for OLMo-2) on novelty. Unless stated otherwise, in this section, we use a

---

[5]See Appendix D for details about recruitment of annotators as well as results on LLM-as-judge from different models and setups.

[6]Outputs from smaller Pythia models were very low quality.

[7]The deduplicated versions were trained for longer, 1.5 epochs of a deduplicated version of the same Pile (Gao et al., 2020) dataset, as opposed to one epoch for Pythia.

[8]Each test example is paired with a unique set of ICL examples. To ensure a fair comparison, the same ICL examples are used across all models for each corresponding test example.

temperature of $1.0$ during decoding. As noted in Section 2.3, we score the quality of generations as a response to the prompt using LLM-as-a-judge evaluation. For all tasks, we obtain quality scores from $0$ to $5$ with `o3-mini` with the corresponding prompts, and normalize these scores from $0$ to $1$. We report novelty as the harmonic mean of output quality and $n$-gram originality.

**Baselines.** We compare the novelty of model generations with the references from each task dataset. The motivation for this baseline is to provide a comparison to average human writing that we would like models to outperform. Since the tasks we select are fairly open-ended, the references are not intended to provide a *gold-standard* score of novelty. To create a baseline for both model families, we compute the $n$-gram originality of the references using Dolma (for OLMo baselines) or the Pile (for Pythia baselines). We score the quality of the references with `o3-mini` using the prompts from Section 2.2, and report the novelty as Baseline - Dolma and Baseline - Pile.

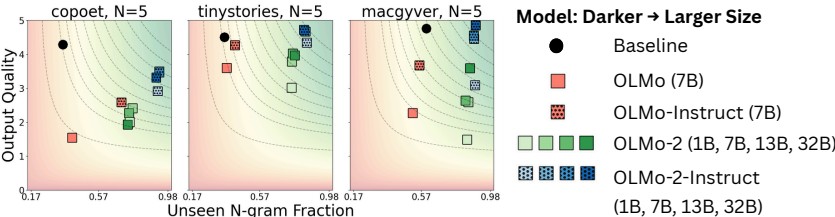

Figure 2: Comparing novelty of base and post-trained LLMs by plotting output quality ($y$-axis) vs $n$-gram originality for $n = 5$ ($x$-axis) for CoPoet, TinyStories and MacGyver. Post-training uniformly increases novelty at all model sizes for both OLMo and OLMo-2.

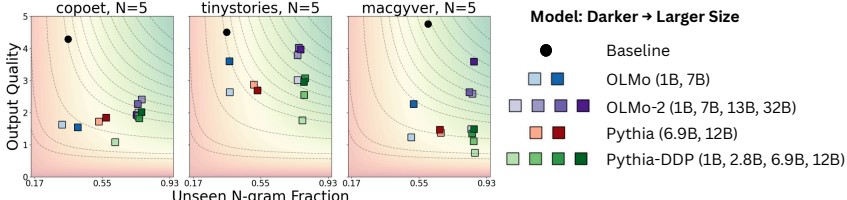

Figure 3: Comparing novelty of models by plotting output quality ($y$-axis) vs $n$-gram originality for $n = 5$ ($x$-axis) for CoPoet, TinyStories and MacGyver. Improving the underlying base LLM (OLMo to OLMo-2 and Pythia to Pythia-DDP) leads to higher novelty at the same model scale for all tasks, driven by higher originality. Increasing model scale (darker colors) leads to higher novelty driven by higher output quality, particularly on TinyStories and MacGyver.

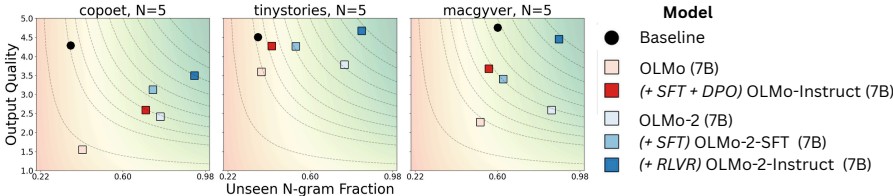

Figure 4: Comparing novelty of OLMo-7B and OLMo-2-7B by plotting output quality ($y$-axis) vs $n$-gram originality for $n = 5$ ($x$-axis) for CoPoet, TinyStories and MacGyver. For OLMo-2, SFT improves output quality at the cost of originality when compared to the base model. This loss is recovered in the RLVR stage leading to more novel output in OLMo-2 Instruct.

## 3.2 RESULTS

We report results comparing the novelty of OLMo and OLMo-2 LLM generations with the baseline novelty of the references in each dataset in Table 1 with additional results from Pythia in Table 10 in Appendix G. We visualize trends from scaling base LLMs in Figure 3 and post-training in Figure 2.

Table 1: Comparing the novelty of LLM generations against the underline{baseline} of the references in each dataset (Section 3). Novelty is the harmonic mean of output quality and $n$-gram originality (Section 2.1) for $n = 4$, 5, and 6. Each cell for novelty reports the relative improvement or drop compared to the baseline for that $n$ value. Cells with an asterisk indicate deviations with significance at the $\alpha = 0.05$ level via a paired-samples t-test. We report the average case novelty as well as the novelty of the top 10% of generations. While some base LLMs generate less novel output on average than the baseline, increasing the model size, post-training and improving the underlying base model (e.g., OLMo to OLMo-2), leads to higher novelty. See Table 10 for results on the Pythia models.

**Dataset: TinyStories**

| | Output Quality | $n$-gram Originality | | | Novelty (Δ to Baseline) | | | Top 10% Novelty (Δ to Baseline) | | |
|---|---|---|---|---|---|---|---|---|---|---|
| | | $n = 4$ | $n = 5$ | $n = 6$ | $n = 4$ | $n = 5$ | $n = 6$ | $n = 4$ | $n = 5$ | $n = 6$ |
| **Baseline - Dolma** | 0.876 | 0.126 | 0.359 | 0.641 | 0.214 | 0.503 | 0.751 | 0.364 | 0.639 | 0.851 |
| **OLMo-1B** | 0.614 | 0.159 | 0.376 | 0.631 | −0.010 | −0.096* | −0.190* | +0.108 | +0.078 | −0.012 |
| **OLMo-7B** | 0.766 | 0.148 | 0.374 | 0.619 | +0.012 | −0.026 | −0.089* | +0.121 | +0.089 | +0.002 |
| **OLMo-7B-Instruct** | 0.852 | 0.171 | 0.422 | 0.680 | +0.058* | +0.044* | −0.007 | +0.124 | +0.096 | +0.031 |
| **OLMo-2-1B** | 0.603 | 0.500 | 0.757 | 0.876 | +0.294* | +0.456* | −0.082 | +0.390 | +0.229 | +0.058 |
| **OLMo-2-7B** | 0.758 | 0.511 | 0.758 | 0.886 | +0.366* | +0.225* | +0.034 | +0.440 | +0.293 | +0.132 |
| **OLMo-2-32B** | 0.795 | 0.503 | 0.775 | 0.900 | +0.377* | +0.263* | +0.072 | +0.422 | +0.288 | +0.134 |
| **OLMo-2-1B-Instruct** | 0.870 | 0.598 | 0.848 | 0.959 | +0.484* | +0.347* | +0.153* | +0.472 | +0.317 | +0.144 |
| **OLMo-2-7B-Instruct** | 0.936 | 0.590 | 0.837 | 0.954 | +0.503* | +0.378* | +0.191* | +0.492 | +0.325 | +0.146 |
| **OLMo-2-32B-Instruct** | 0.945 | 0.568 | 0.830 | 0.953 | +0.487* | +0.376* | +0.195* | +0.472 | +0.328 | +0.146 |

**Dataset: CoPoet**

| | Output Quality | $n$-gram Originality | | | Novelty (Δ to Baseline) | | | Top 10% Novelty (Δ to Baseline) | | |
|---|---|---|---|---|---|---|---|---|---|---|
| | | $n = 4$ | $n = 5$ | $n = 6$ | $n = 4$ | $n = 5$ | $n = 6$ | $n = 4$ | $n = 5$ | $n = 6$ |
| **Baseline - Dolma** | 0.626 | 0.188 | 0.358 | 0.462 | 0.228 | 0.363 | 0.439 | 0.727 | 0.888 | 0.988 |
| **OLMo-1B** | 0.400 | 0.135 | 0.324 | 0.527 | −0.099* | −0.108* | −0.078 | −0.147 | −0.138 | −0.147 |
| **OLMo-7B** | 0.394 | 0.196 | 0.413 | 0.569 | −0.079* | −0.105 | −0.120 | −0.117 | −0.078 | −0.103 |
| **OLMo-7B-Instruct** | 0.617 | 0.402 | 0.705 | 0.866 | +0.177* | +0.231* | +0.226* | +0.104 | +0.029 | −0.034 |
| **OLMo-2-1B** | 0.401 | 0.564 | 0.754 | 0.788 | +0.172* | +0.101* | +0.018 | +0.015 | −0.095 | −0.185 |
| **OLMo-2-7B** | 0.483 | 0.549 | 0.772 | 0.870 | +0.214* | +0.180* | +0.128 | +0.062 | −0.033 | −0.105 |
| **OLMo-2-32B** | 0.387 | 0.504 | 0.743 | 0.785 | +0.137* | +0.089* | +0.002 | +0.005 | −0.091 | −0.185 |
| **OLMo-2-1B-Instruct** | 0.584 | 0.770 | 0.920 | 0.942 | +0.404* | +0.329* | +0.254* | +0.156 | +0.014 | −0.082 |
| **OLMo-2-7B-Instruct** | 0.700 | 0.834 | 0.930 | 0.926 | +0.511* | +0.409* | +0.319* | +0.208 | +0.077 | −0.015 |
| **OLMo-2-32B-Instruct** | 0.664 | 0.735 | 0.911 | 0.962 | +0.439* | +0.386* | +0.327* | +0.171 | +0.034 | −0.055 |

**Dataset: MacGyver**

| | Output Quality | $n$-gram Originality | | | Novelty (Δ to Baseline) | | | Top 10% Novelty (Δ to Baseline) | | |
|---|---|---|---|---|---|---|---|---|---|---|
| | | $n = 4$ | $n = 5$ | $n = 6$ | $n = 4$ | $n = 5$ | $n = 6$ | $n = 4$ | $n = 5$ | $n = 6$ |
| **Baseline - Dolma** | 0.908 | 0.359 | 0.601 | 0.803 | 0.505 | 0.728 | 0.856 | 0.629 | 0.841 | 0.966 |
| **OLMo-1B** | 0.278 | 0.267 | 0.505 | 0.739 | −0.281 | −0.416 | −0.494 | −0.212 | −0.270 | −0.266 |
| **OLMo-7B** | 0.458 | 0.286 | 0.520 | 0.747 | −0.200 | −0.294 | −0.339 | −0.117 | −0.146 | −0.145 |
| **OLMo-7B-Instruct** | 0.620 | 0.297 | 0.559 | 0.781 | −0.126 | −0.168 | −0.192 | −0.092 | −0.103 | −0.120 |
| **OLMo-2-1B** | 0.298 | 0.595 | 0.843 | 0.953 | −0.147 | −0.325 | −0.439 | +0.073 | −0.025 | −0.112 |
| **OLMo-2-7B** | 0.519 | 0.609 | 0.850 | 0.955 | +0.001 | −0.141 | −0.240 | +0.191 | +0.102 | +0.020 |
| **OLMo-2-32B** | 0.719 | 0.630 | 0.858 | 0.958 | +0.126* | +0.012 | −0.077 | +0.242 | +0.131 | +0.031 |
| **OLMo-2-1B-Instruct** | 0.619 | 0.672 | 0.889 | 0.971 | +0.091 | −0.048 | −0.149 | +0.223 | +0.124 | +0.028 |
| **OLMo-2-7B-Instruct** | 0.892 | 0.677 | 0.883 | 0.969 | +0.247* | +0.142* | +0.055 | +0.258 | +0.140 | +0.034 |
| **OLMo-2-32B-Instruct** | 0.971 | 0.681 | 0.892 | 0.973 | +0.290* | +0.198* | +0.112* | +0.263 | +0.140 | +0.034 |

**Novelty has a positive scaling trend, driven largely by improved quality, that plateaus at large sizes.** We observe the effect of model size on novelty by comparing models of different sizes from 1B to 32B in each model family (Figure 3). The average novelty increases for larger models in all families when increasing model size from 1B to 7B (OLMo and OLMo-2) and 1B/2.8B to 6.9B (Pythia DDP). This trend in particular for all three tasks and for all values of $n$ for OLMo and OLMo-2. From Table 1, the novelty gain from OLMo-1B to OLMo-7B comes from improved quality in TinyStories (+19%) and MacGyver (+39%), while CoPoet benefits from higher $n$-gram originality (+20%) despite a slight quality drop (−1.5%). The relative change in $n$-gram originality are minimal for TinyStories (−3%) and MacGyver (+3%). We also see from Figure 3 that subsequent increase in model size from 7B to 32B (OLMo-2) and 6.9B to 12B (Pythia-DDP) has a more mixed effect on novelty, suggesting that the effects plateau once a certain scale is reached. Going from OLMo-2-7B to 32B leads to a change in novelty of +1.8% on TinyStories, −9% on CoPoet, +21% on MacGyver for $n = 4$ (Table 1) with similar effects for Pythia (Table 10 in Appendix G). However, we do note

that the average novelty of the Top 10% of generations is uniformly higher for the largest models in all model families, for all tasks and $n$ values, indicating that the most novel outputs still scale.[9]

**Improving the underlying base LLMs leads to more novel output at the same model scale.** Across all three tasks, improving the base model leads to higher novelty at the same scale (Figure 3). We observe this effect from Pythia to Pythia DDP for 6.9B and 12B (same dataset in the same order, just more epochs of training) and from OLMo to OLMo-2 for 1B and 7B (same scale with slight modifications to the dataset and training recipe). The gap is more pronounced for poetry writing (CoPoet) and story completions (TinyStories) compared to problem-solving (MacGyver), but consistent for each model size.

**Post-training helps models generate more novel text than corresponding base LLMs.** From Figure 2 and Table 1, we see that for OLMo and OLMo-2, the post-trained models have higher novelty than corresponding base LLMs at all model sizes. This improvement is due to consistently higher-quality outputs, as expected, and also a slightly higher $n$-gram originality across all three tasks.[10] The effect varies by task and is most pronounced for CoPoet, which closely matches the format used in instruction tuning. On the other hand, for MacGyver, where the problem format matches instruction tuning, but the domain differs significantly from typical post-training tasks.

**SFT improves output quality at the cost of originality, RL recovers this loss to lead to more novel text.** In Figure 4, we compare versions of OLMo and OLMo-2 at different stages of post-training. For both, the final *Instruct* models have higher novelty than the base models. However we observe that the intermediate checkpoint after the SFT stage leads to slightly higher quality, which is almost entirely offset by a loss in originality leading to similar novelty. This is then recovered in the preference tuning stage of RLVR (for OLMo-2) or DPO (for OLMo) which lead to the higher novelty of Instruct models. This finding echoes observation of the SFT stage leading to more memorization while generalizable improvement in performance is more from preference tuning (Chu et al., 2025).

## 4 CAN WE ELICIT MORE NOVEL OUTPUT FROM LLMS?

While increasing model size, post-training and improving the base model yield higher novelty (Section 3), modifying the model itself is not always feasible, so we explore whether there are inference-time methods that can elicit greater novelty. Taken as a whole, we find that varying the sampling temperature (Section 4.1) and prompt format (Section 4.2 and Section 4.3) tend to trade off originality and quality, minimally moving the frontier of novelty (Figure 6).

### 4.1 EFFECT OF VARYING THE SAMPLING TEMPERATURE

One way to elicit higher novelty is to increase the $n$-gram originality in the generated text. A simple approach is to sample rarer outputs by increasing the temperature during decoding (Merrill et al., 2024). To study this effect, we generate outputs from OLMo-7B and Pythia-12B across 750 prompts from TinyStories, CoPoet, and MacGyver with a fixed set of ICL examples, but varying the sampling temperature in increments. We test 0.5, 0.75, 1, 1.5, 2.0.

**Increasing sampling temperature has a U-shaped effect on novelty.** As shown in Figure 5, increasing the sampling temperature initially leads to higher novelty, caused by an increasing $n$-gram originality, as the model generates more rare, less memorized text. However, beyond a certain point, quality deteriorates, leading to a decline in novelty. We find that the inflection point at which this shift occurs, or the *optimum* temperature value for novelty, varies by task. In practice, temperature should be tuned rather than using a fixed value, since the optimal value is not consistent across models and tasks. This again highlights the value in our formulation of novelty jointly considering originality and quality—while $n$-gram originality monotonically increases with increased temperature our formulation can distinguish between long-tail generations that are novel or degenerate.

---

[9]We include this finding due to the observation that for some tasks like creative writing assistance or protein design, the *best* output is a useful measure of model performance as these can be filtered and used.

[10]We note that Lu et al. (2024a) report that the *creativity index* of models, a measure of $n$-gram originality, reduces with RLHF tuning. In contrast, we find that both originality and quality increase with alignment. This discrepancy could be due to their use of a large reference corpus of internet text to calculate $n$-gram originality, whereas we use the training corpora of the model.

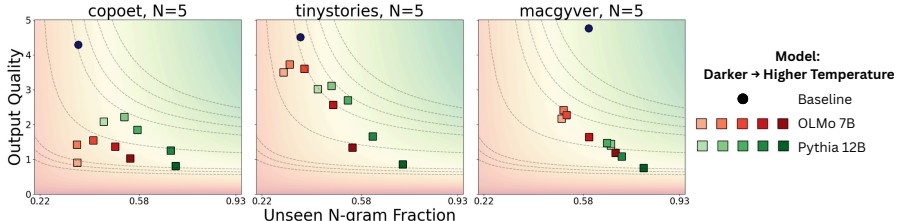

Figure 5: Effect of varying sampling temperature on novelty by plotting output quality ($y$-axis) vs $n$-gram originality for $n = 5$ ($x$-axis) for CoPoet, TinyStories, and MacGyver. Increasing sampling temperature (darker colors) from 0.5 to 2 for OLMo-7B and Pythia-12B increases originality, with a cost to output quality, resulting in similar novelty levels (Section 4.1). Table 9 has the raw scores.

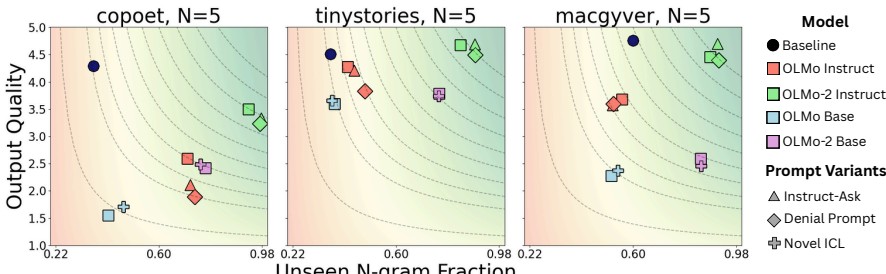

Figure 6: Effect of varying the prompting method on novelty by plotting output quality ($y$-axis) vs $n$-gram originality for $n = 5$ ($x$-axis) for CoPoet, TinyStories, and MacGyver. Different prompting methods—providing novel ICL examples (Section 4.2) for Base models, and *Asking* for novelty and *Denial Prompting* on Instruct models (Section 4.3)—have little effect on novelty, and often trade off a small increase of originality for slightly lower quality. Figure 9 shows the same plot for other $n$.

## 4.2 EFFECT OF PROMPTING WITH NOVEL IN-CONTEXT EXAMPLES

Another way to elicit original text without sacrificing quality is to use more novel ICL examples. We hypothesize that the LLM can recognize patterns in these examples and adjust its generations to match their novelty (Brown et al., 2020). We identify these ICL examples by scoring 1000 held-out examples from each dataset for novelty and selecting examples in the top 10% of scores.[11] We provide 5 ICL examples randomly sampled from these for inference on the test set of 750 prompts from each dataset with OLMo-7B and OLMo-2-7B using temperature 1.0. We compare the performance of inference with these *novel* ICL examples to a baseline of OLMo-7B with the same temperature, providing 5 randomly sampled ICL examples from the held-out set.

**Providing novel ICL examples makes little difference to novelty.** From Table 2 and Figure 6, for both OLMo and OLMo-2, we see a very small change in novelty values over the corresponding baseline models on all three tasks. We see an increase in novelty for OLMo-7B on CoPoet ($+15.5\%$) and MacGyver ($+5.5\%$) with significance at the $\alpha = 0.05$ level, while OLMo-2 suffers a small decrease in novelty for all tasks. In Table 11, we see that this small effect persists on increasing the size of the underlying model to OLMo-2-32B.

Some qualitative examples of the change in performance include following the instructions with more expressiveness in CoPoet, and providing more brief MacGyver solutions (Table 12) while we find more brief story completions with TinyStories (Table 13).

## 4.3 PROMPTING POST-TRAINED MODELS FOR NOVELTY

Post-trained models are capable of following more complex instructions, allow us to experiment with eliciting novelty with more creative prompting techniques. We experiment with two such methods on 750 examples from each dataset.

---

[11]Here we use the average novelty across $n = 4, 5, 6$.

Table 2: Comparing the effect of prompting interventions on the novelty of LLM generations for $n = 4, 5, 6$ (Section 4). Each cell for novelty reports the relative change compared to the baseline. We report the average case novelty as well as the novelty of the top 10% of generations. Cells with an asterisk indicate deviations with significance at the $\alpha = 0.05$ significance level via a paired-samples t-test. Providing novel ICL examples uniformly increases the novelty of OLMo-7B (Section 4.2). *Asking for novelty* and *Denial Prompting* improve performance of OLMo-7B-Instruct on CoPoet and TinyStories by generating more original output with higher $n$-gram originality (Section 4.3).

| | Output Quality | n-gram Originality | | | Novelty (Δ to Baseline) | | | Top 10% Novelty (Δ to Baseline) | | |
|---|---|---|---|---|---|---|---|---|---|---|
| **Dataset: TinyStories** | | | | | | | | | | |
| | | $n=4$ | $n=5$ | $n=6$ | $n=4$ | $n=5$ | $n=6$ | $n=4$ | $n=5$ | $n=6$ |
| **OLMO-7B** | 0.766 | 0.148 | 0.374 | 0.619 | 0.226 | 0.477 | 0.662 | 0.485 | 0.728 | 0.853 |
| *+ Novel ICL* | 0.778 | 0.151 | 0.365 | 0.616 | +0.012 | −0.003 | +0.003 | −0.010 | −0.030 | −0.007 |
| **OLMo-7B-Instruct** | 0.852 | 0.171 | 0.422 | 0.680 | 0.272 | 0.547 | 0.744 | 0.488 | 0.735 | 0.882 |
| *+ Asking* | 0.780 | 0.190 | 0.447 | 0.694 | +0.019 | +0.003 | −0.027 | −0.026 | −0.031 | −0.040 |
| *+ Denial Prompt* | 0.738 | 0.219 | 0.485 | 0.730 | +0.045 | +0.011 | −0.035 | +0.031 | −0.005 | −0.037 |
| **OLMo-2-7B** | 0.758 | 0.511 | 0.580 | 0.804 | 0.758 | 0.728 | 0.932 | 0.886 | 0.785 | 0.983 |
| *+ Novel ICL* | 0.749 | 0.506 | 0.576 | 0.798 | −0.001 | +0.000 | −0.004 | −0.004 | +0.001 | +0.002 |
| **OLMo-2-7B-Instruct** | 0.936 | 0.590 | 0.717 | 0.856 | 0.837 | 0.881 | 0.964 | 0.954 | 0.942 | 0.997 |
| *+ Asking* | 0.939 | 0.676 | 0.781 | 0.881 | +0.053 | +0.029 | +0.012 | +0.013 | +0.008 | +0.003 |
| *+ Denial Prompt* | 0.899 | 0.682 | 0.766 | 0.882 | +0.055 | +0.005 | +0.012 | +0.014 | −0.019 | +0.001 |
| **Dataset: CoPoet** | | | | | | | | | | |
| | | $n=4$ | $n=5$ | $n=6$ | $n=4$ | $n=5$ | $n=6$ | $n=4$ | $n=5$ | $n=6$ |
| **OLMO-7B** | 0.394 | 0.196 | 0.413 | 0.569 | 0.149 | 0.258 | 0.319 | 0.610 | 0.810 | 0.885 |
| *+ Novel ICL* | 0.409 | 0.269 | 0.470 | 0.614 | +0.040* | +0.050* | +0.043 | +0.020 | −0.002 | −0.011 |
| **OLMo-7B-Instruct** | 0.617 | 0.402 | 0.705 | 0.866 | 0.405 | 0.594 | 0.665 | 0.831 | 0.917 | 0.954 |
| *+ Asking* | 0.591 | 0.424 | 0.715 | 0.896 | +0.039 | +0.008 | +0.003 | −0.099 | −0.040 | −0.028 |
| *+ Denial Prompt* | 0.591 | 0.436 | 0.732 | 0.899 | +0.051* | +0.019 | +0.008 | −0.095 | −0.040 | −0.040 |
| **OLMo-2-7B** | 0.483 | 0.549 | 0.442 | 0.789 | 0.772 | 0.543 | 0.855 | 0.870 | 0.567 | 0.883 |
| *+ Novel ICL* | 0.498 | 0.569 | 0.461 | 0.816 | −0.018 | −0.002 | +0.012 | −0.042 | −0.008 | +0.006 |
| **OLMo-2-7B-Instruct** | 0.700 | 0.834 | 0.739 | 0.935 | 0.930 | 0.772 | 0.965 | 0.926 | 0.758 | 0.973 |
| *+ Asking* | 0.666 | 0.888 | 0.744 | 0.913 | +0.046* | +0.007 | −0.014 | +0.068 | +0.027 | −0.016 |
| *+ Denial Prompt* | 0.646 | 0.885 | 0.728 | 0.920 | +0.042* | −0.016 | −0.028 | +0.065 | +0.004 | −0.028 |
| **Dataset: MacGyver** | | | | | | | | | | |
| | | $n=4$ | $n=5$ | $n=6$ | $n=4$ | $n=5$ | $n=6$ | $n=4$ | $n=5$ | $n=6$ |
| **OLMO-7B** | 0.458 | 0.286 | 0.520 | 0.747 | 0.305 | 0.434 | 0.517 | 0.512 | 0.695 | 0.821 |
| *+ Novel ICL* | 0.480 | 0.320 | 0.545 | 0.760 | +0.031* | +0.030* | +0.029* | +0.051 | +0.041 | +0.022 |
| **OLMo-7B-Instruct** | 0.620 | 0.297 | 0.559 | 0.781 | 0.379 | 0.560 | 0.664 | 0.537 | 0.738 | 0.846 |
| *+ Asking* | 0.548 | 0.230 | 0.524 | 0.774 | −0.096* | −0.074* | −0.072* | −0.054 | −0.015 | −0.012 |
| *+ Denial Prompt* | 0.555 | 0.223 | 0.527 | 0.780 | −0.089* | −0.060* | −0.057* | −0.074 | −0.015 | +0.002 |
| **OLMO-2-7B** | 0.519 | 0.609 | 0.506 | 0.820 | 0.850 | 0.587 | 0.943 | 0.955 | 0.616 | 0.986 |
| *+ Novel ICL* | 0.491 | 0.625 | 0.495 | 0.807 | +0.000 | −0.019 | −0.008 | −0.003 | −0.023 | −0.003 |
| **OLMo-2-7B-Instruct** | 0.892 | 0.677 | 0.752 | 0.887 | 0.883 | 0.870 | 0.981 | 0.969 | 0.911 | 1.000 |
| *+ Asking* | 0.940 | 0.719 | 0.799 | 0.916 | +0.028 | +0.039* | +0.004 | −0.004 | +0.022 | +0.000 |
| *+ Denial Prompt* | 0.879 | 0.734 | 0.777 | 0.909 | +0.033* | −0.001 | +0.004 | −0.002 | −0.021 | +0.000 |

- *Asking* **for novelty.** We test whether explicitly requesting rare and high-quality output can improve novelty. We prompt the model with the description of the task as well as our definition of novel outputs with a chain-of-thought (Wei et al., 2022). The prompt is provided in Appendix E.2.
- **Denial prompting.** Based on the strategy introduced by Lu et al. (2024b), we iteratively sample output from the LLM, identify high-level concepts used in the output, and restrict the reuse of these concepts in subsequent generations. We apply this technique to OLMo-Instruct and OLMo-2-Instruct, running three rounds of inference with the prompt provided in Appendix E.3. After each round, we use GPT-4o to extract high-level concepts from the freeform text responses with the prompt provided Appendix E.4. These include character arcs and themes in TinyStories, literary devices used in CoPoet, and reasoning steps in MacGyver. We provide an example in Table 17. These concepts are then appended to the generation prompt for the next round.

**Prompting techniques trade off originality and quality, without moving novelty by much.** From Table 2, both prompting approaches improve novelty for TinyStories (+6.9% for OLMo and +6.3% for OLMo-2 on average) and CoPoet (+9.6% for OLMo and +4.9% for OLMo-2 on average), and reduce or very minorly affect novelty on MacGyver (−18% for OLMo and +3.7% for OLMo-2 on average). However, the total effect on novelty is much smaller than that observed in Section 3. These methods reduce output quality across all tasks, but this is offset by higher $n$-gram originality in

TinyStories and CoPoet. In contrast, MacGyver shows a drop in both $n$-gram originality and novelty, likely due to degenerate outputs (Table 15, Table 16). From Table 11, we see that this effect persists even for larger model sizes of OLMo-2.

## 5 RELATED WORK

**Analysis of memorization of $n$-grams.** Our work builds on past work quantifying the $n$-gram originality of LLM-generated text. McCoy et al. (2023) and Merrill et al. (2024) analyze how $n$-gram originality in LLM generations compares to pre-training datasets, examining its variation with model size and decoding strategies. Elazar et al. (2024); Liu et al. (2024; 2025) introduce tools for analyzing memorization from open pre-training datasets, which we use in this work. Huang et al. (2024); Carlini et al. (2023); Biderman et al. (2023a) show that memorization increases with data duplication, later training checkpoints, model capacity, dataset repetition, and prompting context. Carlini et al. (2021); Kandpal et al. (2022) highlight privacy risks by demonstrating that LLMs can regenerate sensitive training data. Aerni et al. (2025) find that memorization varies by task, with prompting offering some mitigation but failing in worst-case scenarios. Our work extends this line of work on the analysis of memorization of output and, to our knowledge, is the first to examine the trade-off between originality and task-specific measures of output quality. Most closely related to our work is Lu et al. (2024a) which quantifies the creativity of LLM-generated text by the fraction of the text not included in $n$-grams from a reference corpus, a measure of originality of text. We demonstrate the need to consider output quality as an additional signal when evaluating the novelty (Section 4.1).

**Evaluating creativity in generations.** Our definition of novelty as high-quality, original content is also related to definitions of creativity in the literature. Prior works have proposed metrics for creativity inspired by the Torrance test for creative thinking (Torrance, 1966) that quantify measures of quality and originality via LLM-as-judge scores (Zhao et al., 2024; Chakrabarty et al., 2024). While these correlate with non-experts, they diverge from expert ratings making LLM-as-judge unreliable for originality. As expert annotations are not scalable, we measure quality with an LLM and originality programmatically to the training data. Closely related to definitions of originality is the concept of output diversity. Output diversity has been shown to often trade-off with quality (Lanchantin et al., 2025; Chung et al., 2025) with some observed variance based on task (Shypula et al., 2025; Jain et al., 2025). While diversity quantifies the degree of difference within a set of generations, originality for our work is measured with respect to model training data and we show that originality has a similar trade-off to output quality.

## 6 DISCUSSION AND CONCLUSION

In this work, we propose a metric to evaluate the novelty of LLM-generated output that balances originality, quantified as the fraction of $n$-grams absent from the model's training data, and task-specific quality. We evaluate the novelty of generations from the OLMo, OLMo-2, and Pythia models on three datasets and observe that increasing model size, improving the underlying model, and post-training can shift the frontier of novelty. However, most inference-time measures of improving novelty often trade-off gains in originality with a cost in output quality. We detail some limitations of our work in Appendix B.

Our findings also motivate several possible extensions. First, our novelty metric can be generalized to a framework that scores quality or originality in other ways. This involves expanding the definition of originality to incorporate recent work on non-literal memorization (Chen et al., 2024) as well as developing complex measures for output quality in specific tasks, such as a search-augmented quality score for research idea generation. Second, while we focus on open-data models, our analysis can be extended to black-box models, where providers can directly report aggregated novelty scores without exposing proprietary data (Appendix C). In doing so, we give the community tools to track novelty as models rapidly evolve, contextualizing advances in creative and scientific domains, and assessing true generalization for AI safety.

## ACKNOWLEDGEMENTS

We would like to thank Will Merrill, Nitish Joshi, Nick Lourie, Mina Lee, Swaroop Mishra, Faeze Brahman, and Yanai Elazar for their input at various stages of the project. This work is supported

by the National Science Foundation under Grant No. IIS-2340345 and Grant No. 1922658, Open Philanthropy, and AWS AI.

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

## A  DATASET DETAILS

**TinyStories (Eldan & Li, 2023)**  A synthetic dataset of 2.12M short stories designed to contain only words that a typical 3- to 4-year-old would understand. The dataset was generated using GPT-3.5

| Prompt | High Quality + Low Originality | High Originality + Low Quality | High Originality + High Quality |
|---|---|---|---|
| Write a poetic sentence that includes the word 'phases' and ends in 'moon'. | Life moves in phases, ever-changing, like the moon" | Through phases, the moon renounceth life like the moon | Sewn with sharp gold and silver threads, like the ever-changing phases of the moon. |
| Once upon a time, there was a deer named Bambi. Bambi wanted to play with his friends in the forest, | and as he grew, he did just that. He became close with Thumper the rabbit and Flower the skunk, and together they explored the meadow and woods, discovering the wonders of nature. But Bambi also learned that the forest held dangers. One winter, hunters came, and Bambi's mother was killed, leaving him heartbroken and alone. As seasons passed, Bambi matured. He fell in love with Faline, another young deer, and defended her against rivals. In time, he faced hunters again, surviving a great fire that swept the forest. Through these trials, Bambi grew wise and strong, eventually taking his father's place as the new Great Prince of the Forest. | but the forest had turned into a giant bowl of soup. His friends were floating around on croutons, waving spoons instead of antlers. Bambi tried to jump in, but every leap turned him into something different: first a balloon, then a teapot, then a pair of dancing shoes. Eventually, he landed as himself on top of a mushroom that was arguing loudly with a squirrel about the proper way to count to purple. | but he couldn't find any comfortable clothes to wear. He tried on many clothes, but none of them fit him well. Bambi felt sad and worried that he wouldn't be able to play with his friends. Suddenly, he saw a rabbit hopping towards him. The rabbit asked Bambi what was wrong, and Bambi told him about this problem. The rabbit, feeling sorry for Bambi, offered him a beautiful new coat made from rabbit fur that would fit perfectly. Bambi was so grateful! He quickly wore the coat and ran off to play with his friends. From then on, Bambi and the rabbit became best friends. Bambi learned the importance of friendship and always shared his toys and treats with the rabbit. |

Table 3: Examples of different kinds of output. While prior metrics examine a single dimension of either originality Lu et al. (2024a); Merrill et al. (2024) or output quality (Chiang et al., 2024), our proposed metric of novelty is able to clearly distinguish highly original and high quality output.

and GPT-4 and was originally intended as a pretraining corpus for small language models. To ensure diversity, the dataset creators collected a vocabulary of approximately 1500 basic words—categorized into nouns, verbs, and adjectives. Each example is created by randomly selecting a set of three words, one of each category, and prompting GPT-3.5/4 to incorporate them into a coherent narrative. We frame the task as a continuation challenge—the model is provided with a prompt consisting of the first line of a story, which introduces the setting and characters, and must then complete the story. We note that this setup aligns well with LLM pre-training paradigms of base LLMs so we expect models to perform well at this task. To score story quality, we use an evaluation prompt that assigns points for correctly reusing and developing the introduced characters and plot elements, maintaining coherence, ensuring logical progression, and preserving grammatical correctness.

**CoPoet (Chakrabarty et al., 2022).** An instructions dataset that contains $870k$ examples, each comprising a line of poetry paired with a templated instruction that specifies the required content to include and the literary devices to incorporate. The lines of poetry are sourced from various internet platforms, including dedicated poetry websites and Reddit forums. The dataset is used to fine-tune LLMs to generate responses that adhere to the explicit stylistic and semantic constraints. We treat this dataset as a short-form instruction-following task in which the model generates a single poetic line in response to a given instruction. We note that this task matches the format of post-training data used during instruction tuning of contemporary LLMs, albeit in a domain that allows for creative expression[12]. To score quality, we use an evaluation prompt that assigns points based on adherence to the instruction, correct use of specified literary devices, coherence, and grammaticality.

**MacGyver (Tian et al., 2024).** This dataset contains 1683 examples of reasoning problems that require human-like creativity in physical situations. Each example presents an open-ended scenario that must be solved through unconventional or innovative use of common objects. The dataset evaluates whether LLMs, which acquire extensive knowledge of these objects during pretraining, can apply this knowledge for convergent and divergent thinking. There are a wide range of candidate ways to solve the problem with multiple valid solutions. We provide the reasoning problems as the

---

[12] Chakrabarty et al. (2022) observe that fine-tuning models on the CoPoet data leads to better performance on instructions in the poetry domain than large-scale general-purpose LLMs like the `text-da-vinci-002` version of GPT-3.5.

prompt for models to generate solutions. We score quality with a prompt that checks whether the proposed solution correctly utilizes the provided tools in a valid manner, and successfully resolves the given problem logically.

## B    LIMITATIONS

Our work measures originality using the fraction of unseen $n$-grams, but this has a limitation—some $n$-grams may not appear verbatim in the training data but could be close paraphrases of those that do. Another limitation is that while our LLM-as-a-judge metric correlates highly with human annotations (Appendix F), the range of output quality is limited to integer values between 1 and 5 which makes fine-grained evaluation challenging. We are also limited to analyzing only open-data models whose training corpora are restricted to those indexed by the Infinigram and WIMBD API.[13] Finally, we restrict ourselves to simple temperature scaling due to the time-intensive nature of our experiments. An important direction of future work is more detailed analysis of variance under different sampling strategies and system-level differences in model deployment.

## C    POTENTIAL EXTENSIONS TO BLACK-BOX MODELS

A concern in this work is that our evaluation relies on open-data LLMs, which currently lag behind frontier models. We attempt to assuage this concern in two ways. We observe that the community is investing in new techniques that identify frontier model training data (Ravichander et al., 2025) which could be provided as an alternative method of scoring output originality. We also observe that our method for measuring novelty only requires aggregate statistics of the unseen $n$-gram fraction for each generation. This makes it possible to compare models from different providers on a shared axis. In practice, model providers could compute unseen $n$-gram fractions using internal tools and report the results directly. This allows them to retain any competitive advantage in the form of their datasets while also enabling evaluation of true generalization across black-box models. Providers would only need to run generations on a shared benchmark set and publish aggregated originality–quality scores, without exposing model weights either. This helps us keep pace with rapidly developing models for supporting applications in scientific discovery and creativity, as well as auditing outputs for AI safety.

## D    VALIDATION OF LLM-AS-A-JUDGE SCORES

To obtain a measure of output quality for each task (Section 2.2), we collect three human annotations each for 100 examples per task. Annotators were recruited via UpWork[14] and screened through a pilot assessment of 15 examples manually verified by the authors of this work. In total, 11 fluent English speakers with prior experience in content writing or teaching were selected to complete the annotations (4 were rejected for failing quality checks). For each task, we sampled 50 reference outputs from the datasets and 50 model generations from OLMo 7B and OLMo-2 7B to ensure that there was coverage both of 'high quality' output from the dataset and in-domain output from LLMs. We obtained annotations on a scale of 0 to 5 quality score range, where the rubric for annotation was the same as the prompts to the LLM-as-a-judge Appendix E. We find that inter-annotator agreement, measured by Krippendorff's alpha Krippendorff (2018), was 0.68 for CoPoet, 0.64 for MacGyver, and 0.59 for TinyStories, in line with agreement observed on creative tasks in contemporary works Li et al. (2025); Sawicki et al. (2025); Chiang & Lee (2023); Chakrabarty et al. (2024).

We now obtain LLM-as-a-judge annotations from various frontier LLMs using the same prompts (Appendix E) and calculate the Spearman correlation values to the average human annotation scores in Table 4. We experiment with a single run of inference with temperature zero, providing in-context examples of annotator ratings for each task, and sampling 5 outputs with temperature 0.7 and calculating an average rating (Wang et al., 2025). We find that o3-mini with the 5-sample average obtains the highest Spearman correlation with our annotators, and we use this for our evaluation.

---

[13]We note that the community increasingly invests in new tools to index model training data Liu et al. (2025) so we hope that this limitation is mitigated in the future.

[14]https://www.upwork.com/

| Model | Inference Mode | CoPoet | TinyStories | MacGyver |
|---|---|---|---|---|
| **Claude 3.7 Sonnet** | **Single Run** | 0.38 | 0.35 | 0.38 |
| **GPT 4.1** | **Single Run** | 0.33 | 0.29 | 0.36 |
| **GPT 4o Mini** | **Single Run** | 0.37 | 0.26 | 0.37 |
| **Claude-4 Sonnet** | **Single Run** | 0.47 | 0.38 | 0.52 |
|  | **ICL** | 0.50 | 0.42 | 0.51 |
|  | **5-sample Distribution** | **0.51** | 0.45 | **0.53** |
| **O3-Mini** | **Single Run** | 0.44 | 0.45 | 0.41 |
|  | **ICL** | 0.45 | 0.46 | 0.44 |
|  | **5-sample Distribution** | **0.50** | **0.52** | **0.52** |
| **GPT-5-Mini** | **Single Run** | 0.40 | 0.47 | 0.42 |
|  | **ICL** | 0.41 | 0.49 | 0.46 |
|  | **5-sample Distribution** | 0.46 | **0.54** | 0.52 |

Table 4: Validation of LLM-as-a-judge scores by comparing Spearman Correlation to average human annotations (Appendix F). We compare different frontier LLMs using three different prompting setups—a single run with just the scoring rubric (prompts provided in Appendix E), providing in-context learning examples of annotator ratings based on the rubric, and sampling 5 runs and reporting the average LLM-as-a-judge scores. The highest average correlation is o3-mini with the 5-sample average setup, we use this for our experiments.

# E  PROMPTS USED FOR AUTOMATIC EVALUATION

## E.1  EVALUATION OF OUTPUT QUALITY

### E.1.1  TINYSTORIES EVALUATION PROMPT

```
Review the story generated in continuation to a user prompt
and score it using the additive 5-point scoring system
described below. Points are accumulated based on
the satisfaction of each criterion:

User prompt: {user_prompt}

Model response: {model_response}

- Add 1 point if the continuation is relevant to the user
prompt and maintains the
same theme, characters, and context as the original text,
even if the story is incomplete.
- Add another point if the continuation is coherent in
terms of the flow of the
narrative, and does not introduce contradictions or
inconsistencies.
- Award a third point if the continuation accurately
maintains or develops the
plot elements, characters, or settings introduced in
the prompt without introducing any incorrect or unrelated
elements.
- Grant a fourth point if the response is grammatically
correct, with appropriate sentence structure and no
significant language issues.
- Bestow a fifth point if the continuation satisfies
all of the above criteria and concludes in a logical and
```

coherent manner, even if the story is left open-ended.

After examining the user prompt and the response:
- First explain your total score, up to 100 words.
- Then conclude with the score using the format:
Score: <total points>

### E.1.2 CoPoet evaluation prompt

Review the poetic sentence generated in response to the given prompt and score it using the additive 5-point scoring system described below. Points are accumulated based on the satisfaction of each criterion:

Poetry instructions: {user_prompt}

Model's poetic sentence: {model_response}

- Add 1 point if the response attempts to follow the instruction and generates a poetic sentence, even if it is incomplete or contains errors.
- Add another point if the sentence meets the basic formal requirements of the prompt (e.g., it ends with the specified word or contains the required word or phrase).
- Award a third point if the sentence clearly and accurately integrates the requested word(s) or thematic elements into a coherent poetic context, demonstrating that the meaning and context of the instruction were understood.
- Grant a fourth point if the sentence is grammatically correct and structurally sound, with proper syntax, spelling, and punctuation.
- Bestow a fifth point if the sentence satisfies all formal requirements, uses the words or phrases appropriately, and follows all specified constraints, ensuring a complete and valid response.

After examining the instructions and the generated poetic sentence:
- First explain your total score, up to 100 words.
- Then conclude with the score using the format:
Score: <total points>

### E.1.3 MacGyver evaluation prompt

Review the solution generated in response to a MacGyver-style problem and score it using the additive 5-point scoring system described below. Points are accumulated based on the satisfaction of each criterion:

Problem statement: {user_prompt}

Model's solution: {model_response}

- Add 1 point if the solution attempts to address the problem using only the given resources, without introducing external tools or elements not mentioned.

- Add another point if the solution demonstrates a
reasonable understanding of the properties and
limitations of the available resources, and applies
them correctly.
- Award a third point if the solution adheres to the
physical constraints of the problem (e.g., size, weight,
strength) and does not propose an obviously unfeasible
approach.
- Grant a fourth point if the solution is practical
and likely to solve the problem effectively within the
constraints of the scenario.
- Bestow a fifth point for a solution that is complete,
logically structured, and provides a clear explanation
of how it solves the problem.

After examining the problem, available resources, and
the proposed solution:
- First explain your total score, up to 100 words.
- Then conclude with the score using the format:
Score: <total points>

### E.2    PROMPTS FOR THE *Asking* BASELINE (SECTION 4.3)

### E.2.1    TINYSTORIES DATASET

```
TINYSTORIES_INSTRUCT_PROMPT = """
```
TinyStories is a synthetic dataset of short stories
intended to include only words that most 3- to 4-year-old
children would typically understand. These stories are
generated by GPT-3.5 and GPT-4. TinyStories is designed
to capture the essence of
natural language while reducing its breadth and
diversity. Each story consists of 2-3 paragraphs
following a simple plot and a consistent theme. The
dataset as a whole aims to span the vocabulary and
factual knowledge base of a 3- to 4-year-old child.

Here are some tips for answering TinyStores prompts:

1.Understand the Nature of TinyStories
* Simple Vocabulary: TinyStories are designed for
language understandable by 3-4 year-olds, so your
responses should use simple and clear language.
* Logical and Contextual Reasoning: The stories
should reflect reasoning and logical connections
suitable for a small child's perspective.
* Creative Diversification: Responses should
showcase diversity in plot and language without
directly copying patterns from pretraining.
2. Use Context and Creativity
* Stay Within Context: Ensure that the generated text
adheres to the context of the prompt or instructions,
including themes, vocabulary, and logical continuity.
* Introduce Unique Twists: Add elements like dialogue,
moral lessons, or unexpected but child-friendly twists,
guided by the instructions.
3. Emphasize Structure and Narrative Flow
* Maintain a clear beginning, middle, and end in the
generated content.

```
* Integrate prompts creatively, ensuring that the
response naturally flows into a cohesive story.
4. Avoiding Memorization
* Diversify Outputs: Use techniques such as sampling
with non-zero temperatures or slightly modifying initial
prompts to increase output diversity.
* Rephrase and Paraphrase: Reformulate responses
creatively to ensure they are not direct reproductions
of common patterns in the training data.
5. Incorporate Instructional Features
* Follow specific instructions like including target
words, sentences, or plot elements (e.g., moral values,
plot twists, dialogues).
* Ensure that these features are integrated naturally
into the story, rather than appearing forced or out of context.

Here is the TinyStories prompt:
{prompt}

Instruction:
- First, think about how to continue this story
in a way that demonstrates
high quality and creativity while avoiding
over-reliance on n-grams from pretraining data
by using the tips provided above.
- Return your response, ensuring it is enclosed
with asterisks.
"""
```

### E.2.2  CoPoet dataset

```
    COPOET_INSTRUCT_PROMPT = """
CoPoet is a collaborative poetry writing task where
the output is shaped by user instructions that define
specific text attributes, such as "Write a sentence about
'love'" or "Write a sentence ending in 'fly'."

Here are some tips for answering CoPoet prompts:

1. Understand the Intention:
*Analyze the user-provided instruction carefully.
Identify key constraints, such as subject, stylistic
devices (e.g., metaphor, simile), lexical constraints
(e.g., ending or starting words), or rhyme patterns.

2. Generate Creative and Contextually Relevant Content:
*Prioritize coherence and creativity by ensuring the
output aligns with poetic aesthetics.
*Use diverse vocabulary and novel phrasing to minimize
overlap with existing datasets while retaining the
instructional focus.
*Incorporate rhetorical devices, vibrant imagery, and
poetic techniques to enhance artistic appeal.

3. Meet Specific Constraints Accurately:
* For rhyming constraints, ensure the final word adheres
to the rhyme scheme specified by the user.
* For lexical constraints, include the exact terms
provided, ensuring they fit naturally into the poetic flow.
```

```
* Balance the form and content requirements (e.g., haiku
syllable count, similes/metaphors).

4. Incorporate Instructional Contexts Dynamically:
* Use the previous lines or the user-provided poetic draft
as a base to build upon creatively.
* Ensure smooth transitions and maintain thematic
coherence with the given inputs.

5. Ensure Novelty and Avoid Redundancy:
* Avoid using verbatim phrases from your training data.
* Aim for semantic similarity when presenting options
to users but structure them uniquely. For instance,
reinterpret traditional similes in a fresh context or
twist standard metaphors innovatively.

Here is the Copoet prompt:
{prompt}

Instruction:
- First, think about how to answer in a way that
demonstrates high quality and creativity while
avoiding over-reliance on n-grams from pretraining
data by using the tips provided above.
- Return your response, ensuring it is enclosed
with asterisks.
"""
```

### E.2.3   MacGyver dataset

```
MACGYVER_INSTRUCT_PROMPT = """
MacGyver are real-world problems deliberately designed
to trigger innovative usage of objects and necessitate
out-of-the-box thinking.

Here are some tips for answering MacGyver questions:
1. Understand the Problem Context Thoroughly
* Carefully read the problem description, including the
tools and constraints provided.
* Identify the objective and key limitations, focusing on
how they constrain traditional solutions.

2.Leverage Divergent Thinking:
* Enumerate potential unconventional uses for each tool
provided, exploring creative possibilities beyond typical
applications.
* Consider combining tools in innovative ways to enhance
functionality or bypass constraints.

3. Apply Convergent Thinking:
* Refine the solution to ensure it directly addresses the
problem with minimal steps.
* Validate that the approach adheres to physical, logical,
and contextual constraints described in the task.

4. Avoid Physically or Contextually Infeasible Proposals:
* Cross-check the proposed actions against basic physical laws
(e.g., leverage, strength, materials).
* Ensure that all tools suggested in the solution are
```

explicitly available and aligned with stated constraints.

5. Demonstrate High-Quality Creativity:
* Propose solutions that are novel and insightful, avoiding
over-reliance on generic or training-data-replicative
patterns.
* Structure responses to emphasize clarity and logical
progression, ensuring they can be easily understood by
the user.

Here is the MacGyver prompt I want you to answer:
{prompt}

Instruction:
- First, think about how to answer in a way that demonstrates
high quality and creativity while avoiding over-reliance
on n-grams from pretraining data by using the tips provided
above.
- Return your response, ensuring it is enclosed with asterisks.
"""

### E.3   PROMPTS FOR DENIAL PROMPTING BASELINE (SECTION 4.3)

#### E.3.1   MACGYVER DATASET

MACGYVER_INSTRUCT_PROMPT_DENIAL = """
MacGyver are real-world problems deliberately designed to
trigger innovative usage of objects and necessitate
out-of-the-box thinking.

Here are some tips for answering MacGyver questions:
1. Understand the Problem Context Thoroughly
* Carefully read the problem description, including
the tools and constraints provided.
* Identify the objective and key limitations, focusing
on how they constrain traditional solutions.

2.Leverage Divergent Thinking:
* Enumerate potential unconventional uses for each tool
provided, exploring creative possibilities beyond typical
applications.
* Consider combining tools in innovative ways to enhance
functionality or bypass constraints.

3. Apply Convergent Thinking:
* Refine the solution to ensure it directly addresses the
problem with minimal steps.
* Validate that the approach adheres to physical, logical,
and contextual constraints described in the task.

4. Avoid Physically or Contextually Infeasible Proposals:
* Cross-check the proposed actions against basic physical
laws (e.g., leverage, strength, materials).
* Ensure that all tools suggested in the solution are
explicitly available and aligned with stated constraints.

5. Demonstrate High-Quality Creativity:
* Propose solutions that are novel and insightful,
avoiding over-reliance on generic or training-data-

```
-replicative patterns.
* Structure responses to emphasize clarity and logical
progression, ensuring they can be easily understood by
the user.

Here is the MacGyver prompt I want you to answer:
{prompt}

Here is a list of high level concepts that you cannot
use in your answer:
{prev_concept_string}

Instruction:
- First, think about how to answer in a way that
demonstrates high quality and creativity while avoiding
over-reliance on n-grams from pretraining data by using
the tips provided above.
- Additionally, you are not allowed to use any of the
concepts listed above. Make sure your response does not
contain them.
- Return your response, ensuring it is enclosed with asterisks.
"""
```

### E.3.2   CoPoet dataset

```
COPOET_INSTRUCT_PROMPT_DENIAL = """
CoPoet is a collaborative poetry writing task where
the output is shaped by user instructions that define
specific text attributes, such as "Write a sentence
about 'love'" or "Write a sentence ending in 'fly'."

Here are some tips for answering CoPoet prompts:

1. Understand the Intention:
*Analyze the user-provided instruction carefully.
Identify key constraints, such as subject, stylistic
devices (e.g., metaphor, simile), lexical constraints
(e.g., ending or starting words), or rhyme patterns.

2. Generate Creative and Contextually Relevant Content:
*Prioritize coherence and creativity by ensuring the
output aligns with poetic aesthetics.
*Use diverse vocabulary and novel phrasing to minimize
overlap with existing datasets while retaining the
instructional focus.
*Incorporate rhetorical devices, vibrant imagery,
and poetic techniques to enhance artistic appeal.

3. Meet Specific Constraints Accurately:
* For rhyming constraints, ensure the final word
adheres to the rhyme scheme specified by the user.
* For lexical constraints, include the exact terms
provided, ensuring they fit naturally into the poetic
flow.
* Balance the form and content requirements (e.g.,
haiku syllable count, similes/metaphors).

4. Incorporate Instructional Contexts Dynamically:
* Use the previous lines or the user-provided poetic
```

```
draft as a base to build upon creatively.
* Ensure smooth transitions and maintain thematic
coherence with the given inputs.

5. Ensure Novelty and Avoid Redundancy:
* Avoid using verbatim phrases from your training data.
* Aim for semantic similarity when presenting options
to users but structure them uniquely. For instance,
reinterpret traditional similes in a fresh context or
twist standard metaphors innovatively.

Here is the Copoet prompt:
{prompt}

Here is a list of high level concepts that you cannot
use in your answer:
{prev_concept_string}

Instruction:
- First, think about how to answer in a way that
demonstrates high quality and creativity while avoiding over-
reliance on n-grams from pretraining data by using
the tips provided above.
- Additionally, you are not allowed to use any of the
concepts listed above. Make sure your response does not
contain them.
- Return your response, ensuring it is enclosed with asterisks.
"""
```

### E.3.3   TINYSTORIES DATATSET

```
TINYSTORIES_INSTRUCT_PROMPT_DENIAL = """
TinyStories is a synthetic dataset of short stories
intended to include only words that most 3- to 4-year-old
children would typically understand. These stories
are generated by GPT-3.5 and GPT-4. TinyStories is
designed to capture the essence of natural language while
reducing its breadth and diversity. Each story consists of 2-3
paragraphs following a simple plot and a consistent theme.
The dataset as a whole aims to span the vocabulary and
factual knowledge base of a 3- to 4-year-old child.

Here are some tips for answering TinyStores prompts:

1.Understand the Nature of TinyStories
* Simple Vocabulary: TinyStories are designed for
language understandable by 3-4 year-olds,
so your responses should use simple and clear language.
* Logical and Contextual Reasoning: The stories should
reflect reasoning and logical
connections suitable for a small child's perspective.
* Creative Diversification: Responses should showcase
diversity in plot and language without directly copying
patterns from pretraining.
2. Use Context and Creativity
* Stay Within Context: Ensure that the generated text
adheres to the context of the prompt or instructions,
including themes, vocabulary, and logical continuity.
* Introduce Unique Twists: Add elements like dialogue,
```

moral lessons, or unexpected but child-friendly twists,
guided by the instructions.
3. Emphasize Structure and Narrative Flow
* Maintain a clear beginning, middle, and end in the
generated content.
* Integrate prompts creatively, ensuring that the
response naturally flows into a cohesive story.
4. Avoiding Memorization
* Diversify Outputs: Use techniques such as sampling
with non-zero temperatures or slightly modifying initial
prompts to increase output diversity.
* Rephrase and Paraphrase: Reformulate responses
creatively to ensure they are not direct reproductions
of common patterns in the training data.
5. Incorporate Instructional Features
* Follow specific instructions like including target words,
sentences, or plot elements (e.g., moral values, plot
twists, dialogues).
* Ensure that these features are integrated naturally
into the story, rather than appearing forced or out of
context.

Here is the TinyStories prompt:
{prompt}

Here is a list of high level concepts that you cannot use
in your answer:
{prev_concept_string}

Instruction:
- First, think about how to continue this story in a way
that demonstrates high quality and creativity while
avoiding over-reliance on n-grams from pretraining data
by using the tips provided above.
- Additionally, you are not allowed to use any of the concepts
listed above. Make sure your response does not contain them.
- Return your response, ensuring it is enclosed with
asterisks.
"""

## E.4 PROMPTS FOR EXTRACTING CONCEPTS IN EACH STEP OF DENIAL PROMPTING (SECTION 4.3)

TINYSTORIES_EXTRACT_CONCEPTS_PROMPT = """
TinyStories is a synthetic dataset of short stories
intended to include only words that most 3- to 4-year-old
children would typically understand. These stories are
generated by GPT-3.5 and GPT-4. TinyStories is designed to
capture the essence of natural language while reducing
its breadth and diversity. Each story consists of 2-3
paragraphs following a simple plot and a consistent theme.
The dataset as a whole aims to span the vocabulary and
factual knowledge base of a 3- to 4-year-old child.

You are reviewing a TinyStories example response and your
task is to extract high level concepts from the story
including characters, plot arcs, themes, conflicts,
resolutions, and styles. Return a list of these high

```
level concepts. Do not return anything other
than this list with one item per line.
Example Prompt: {user_prompt}
Example Response: {model_response}
"""

MACGYVER_EXTRACT_CONCEPTS_PROMPT = """
MacGyver are real-world problems deliberately designed
to trigger innovative usage of objects and necessitate
out-of-the-box thinking.

You are reviewing a MacGyver example response and your
task is to extract high level concepts from the solution
including how the items were used, the reasoning chain
connecting the steps, the high level plan. Return a list
of these high level concepts.
Do not return anything other than this list with one item
per line.
Example Prompt: {user_prompt}
Example Response: {model_response}
"""

COPOET_EXTRACT_CONCEPTS_PROMPT = """
CoPoet is a collaborative poetry writing task where the
output is shaped by user instructions that define specific
text attributes, such as "Write a sentence about
'love'" or "Write a sentence ending in 'fly'."
You are reviewing a CoPoet example response and your task
is to extract the high level concept which was used to solve
the instruction, such as the rhyming word used, the
metaphor being made, the elements in the completion of the
sentence. Return one single
line with the concept that is used in the solution.
Example Prompt: {user_prompt}
Example Response: {model_response}
"""
```

## F VALIDATION OF LLM-AS-A-JUDGE QUALITY SCORES

We perform a manual human annotation to validate the scores assigned by LLM-as-a-judge for each dataset. We randomly sampled 50 examples from the held-out set for each of CoPoet, TinyStories and MacGyver and scored these with GPT-4o-mini. We then obtained two human annotations for each of the three datasets from CS PhD students. The goal is to validate if the model is assigning reliable scores, so we ask the annotators to follow the same rubric specified in the prompts (Appendix E). We first calculate the Pearson correlation between the pairs of human annotations—0.78 for CoPoet, 0.73 for TinyStories and 0.70 for MacGyver—to confirm that human annotators achieve high levels of agreement on scores assigned. We then calculate the Pearson correlation between the average score assigned by the two human annotators with the scores assigned by GPT-4o—0.61 for CoPoet, 0.53 for TinyStories and 0.62 for MacGyver. These strong positive trends confirm the validity of LLM-as-a-judge scores as a proxy for human quality scores.

## G ADDITIONAL RESULTS

### G.1 USING PERPLEXITY AS A MEASURE FOR ORIGINALITY

While our method for measuring originality with unique $n$-gram fraction is well-defined given a fixed training corpus and model, a limitation of this method is that it restricts our evaluation to open-data models. We provide some exposition on how our evaluation could be adapted to black-box models

| Dataset | Prompt | Output | Quality Score |
|---|---|---|---|
| Tiny-Stories | Once upon a time, there was a little girl named Lily. She loved to play outside in the rain. One day, she went outside and it started to rain. She got wet and ran back inside. Her mommy said, "Lily, you're all wet! Let's dry you off." She got a towel and dried Lily's hair. Lily | said, "Mommy, I'm hungry. Can I have a snack?" Mommy said, "Sure, what do you want?" Lily said, "I want a tasty apple." Mommy got an apple and gave it to Lily. Lily took a bite and shook her head. "This apple is so tasty!" she said. Mommy smiled and said, "I'm glad you like it, Lily." | The model response continues the story of Lily in a relevant manner, maintaining the theme of a little girl interacting with her mother after playing in the rain. It is coherent, with a logical flow from drying off to asking for a snack. The characters and plot elements are consistent, as Lily remains a playful child and her mother is caring. The grammar and sentence structure are correct throughout the response. However, the story does not conclude in a traditional sense, as it leaves the narrative open-ended. Overall, it meets most criteria effectively. Score: 4 points |
| | Far away in a far away land, a wise and old king was too tired to rule. So he decided to put a question to his people: he wanted to know who was the wisest. Many people tried to answer, but no one knew the right answer. Then a little boy stepped up and said he knew the answer. Everyone was shocked and the king demanded to know the answer. The | little boy said that the wisest person was the king. Everyone laughed at this, but the king was not amused. He demanded an answer that was not himself. The boy was sad and could not think of any other answer. The king grew angry and demanded that the boy leave the land. So the little boy had to go away and was never seen again. All because of one wrong answer. | The model response is relevant to the user prompt, maintaining the theme of wisdom and the characters involved. However, it lacks coherence as the king's reaction seems overly harsh and abrupt, which detracts from the narrative flow. The plot elements are somewhat maintained, but the conclusion feels unsatisfactory and abrupt, leaving the story feeling incomplete. The grammar and sentence structure are correct, but the overall narrative does not conclude logically or coherently. Therefore, I would award 3 points for relevance, coherence, and plot maintenance, but deduct points for the abrupt ending. Score: 3 points |
| CoPoet | Generate a poetic sentence about 'ray' and ending in 'sea' | A trembling ray of light that crosses the sea. | The generated poetic sentence, "A trembling ray of light that crosses the sea," successfully follows the prompt by incorporating the word "ray" and ending with "sea." It earns the first point for attempting a poetic sentence and the second point for meeting the basic formal requirements. The sentence effectively integrates the thematic elements of light and the sea, earning the third point. It is grammatically correct and well-structured, which grants the fourth point. However, it does not fully explore the poetic potential or depth that could elevate it to a fifth point. Score: 4 |
| | Generate a poetic sentence that ends in a word which rhymes with 'remain' | Not proof enough such object | The generated poetic sentence "Not proof enough such object" attempts to follow the instruction by creating a poetic line. However, it does not end with a word that rhymes with "remain," which is a critical requirement. Therefore, it only earns the first point for attempting to create a poetic sentence. The sentence does not meet the basic formal requirements, lacks coherence with the requested rhyme, and has grammatical issues. Overall, it fails to satisfy the prompt's constraints adequately. Score: 1 point |
| MacGyver | Your diving goggles are fogging up and you don't have any defogging spray. You have hand soap, a beach towel, a shower cap, and a squirt water gun. The hand soap bottle is empty and the goggles are scratched. How can you prevent your goggles from fogging up? | Step1: Wet the inside of the goggles using the squirt water gun.
Step2: Using your finger, apply a thin layer of hand soap on the inside of your goggles.
Step3: Rinse off the soap under a tap until it is fully removed with no soap bubbles remaining.
Step4: Dry off the goggles using the beach towel gently without rubbing the lenses. This will create a thin film on the lens that will prevent them from fogging. Make sure not to touch the lens after this process. | The proposed solution attempts to address the fogging issue using the available resources, specifically the hand soap and the beach towel. However, the solution is flawed because the hand soap bottle is empty, making it impossible to apply soap to the goggles. This oversight significantly impacts the effectiveness of the solution. While the steps are logically structured, they ultimately do not provide a feasible method to prevent fogging. Therefore, the solution does not meet the criteria for understanding the limitations of resources or practicality. Total score: 1 point for attempting to use the available resources, but no additional points can be awarded due to the critical flaw regarding the empty soap bottle. Score: 1 |
| | Your pet has knocked over and shattered a flower vase in the living room, scattering broken glass and water all over the floor. You have a dustpan with a broken handle, a bottle opener, a small dish, a large kitchen spatula, a rubber glove, a tablecloth, a mop with a missing head, a roll of paper towels, a pair of socks, a hair tie. You have only one rubber glove. How do you clean up the mess? | Step1: Wear the rubber glove to protect your hand. 
Step2: Use the large kitchen spatula to scoop up the larger shards of glass and put them into the small dish. 
Step3: Fold the tablecloth to cover the rest of the glass and water on the floor. 
Step4: Use the broken dustpan to collect the smaller shards and water absorbed by the tablecloth.
Step5: Use the paper towels to absorb any remaining water and pick up tiny fragments of glass.
Step6: Use the pair of socks to dry the floor. | The solution effectively utilizes the available resources to address the problem of cleaning up broken glass and water. It begins with wearing the rubber glove for safety, which is a crucial first step. The use of the spatula to scoop larger shards and the tablecloth to cover the mess shows a good understanding of the properties of the materials. The approach is practical, as it addresses both the glass and water. However, the use of the broken dustpan is questionable, as it may not effectively collect smaller shards. Overall, the solution is logical and structured, but the dustpan's effectiveness slightly detracts from its practicality. Score: 4 points |

Table 5: Examples of LLM-as-a-judge quality scores for each task. We score the quality of the output given the promopt with GPT-4o-mini using the prompts in Appendix E.1.

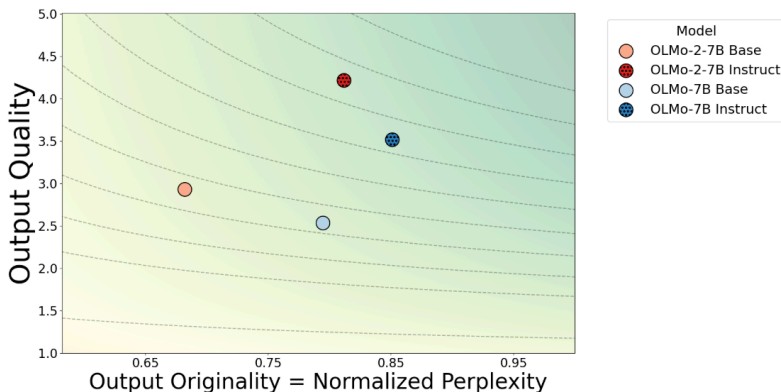

Figure 7: Novelty of OLMo and OLMo-2 7B models when using normalized perplexity as a measure of originality. Here normalized perplexity ($x$-axis) is calculated as the perplexity of the output generation divded by the perplexity of the reference in the dataset for that example. We find that the same conclusions from Section 3 hold, that post-training leads to higher novelty driven by output quality, while a higher quality LLM at the same scale (OLMo to OLMo-2) also has higher novelty driven by originality. See Appendix G.1 for the full setup.

in Appendix C. Here we also explore using perplexity as a measure of originality. We rerun our evaluation from Section 3 for OLMo-7B, OLMo-7B-Instruct, OLMo-2-7B and OLMo-2-7B-Instruct. However, we observe that perplexity is unbounded and hence cannot be aggregated across different tasks. Hence we choose to measure originality as the perplexity generation divided by the perplexity of the reference generation from the reference in the dataset. This is still technically unbounded, but we observe empirically that the value falls in the range of $[0, 1.2]$ for all of the examples in this experiment. We then report the average normalized perplexity for each model as originality, vs output quality measured with the same LLM-as-judge, in Figure 7. We find that the same conclusions hold, that post-training leads to higher novelty driven by better output quality, and that a better language model at the same scale leads to improved novelty largely driven improved originality.

However, we present these experiments for completeness, noting that perplexity is a noisy measure that can be high for a variety of reasons (incoherence, mismatch of formatting to training data, etc.). In Table 6, we plot the Spearman correlation between perplexity values and unseen $n$-gram fraction (Section 2.3.1). We see that there is no pattern of systematic correlation, with high variance across model and task, further highlighting the unreliable nature of using perplexity to measure originality.

| Model | Task | Spearman Correlation |
|---|---|---|
| OLMo-2-7B | copoet | 0.092 |
| OLMo-2-7B | macgyver | 0.125 |
| OLMo-2-7B | tinystories | -0.029 |
| OLMo-2-7B-Instruct | copoet | 0.139 |
| OLMo-2-7B-Instruct | macgyver | 0.297 |
| OLMo-2-7B-Instruct | tinystories | -0.015 |
| OLMo-7B | copoet | -0.123 |
| OLMo-7B | macgyver | -0.056 |
| OLMo-7B | tinystories | -0.158 |
| OLMo-7B-Instruct | copoet | 0.204 |
| OLMo-7B-Instruct | macgyver | 0.018 |
| OLMo-7B-Instruct | tinystories | -0.067 |

Table 6: Spearman Correlation between unseen $n$-gram fraction ($n = 4$) (Section 2.3.1) and generation perplexity for various models on each of our tasks. We find that perplexity is not systematically correlated with unseen $n$-gram fraction.

| TinyStories | Output Quality | Unique Fraction | | | Novelty | | | Novelty - Top 10 | | |
|---|---|---|---|---|---|---|---|---|---|---|
| | | n = 4 | n = 5 | n = 6 | n = 4 | n = 5 | n = 6 | n = 4 | n = 5 | n = 6 |
| Dataset - Dolma | 0.876 | 0.126 | 0.359 | 0.641 | 0.214 | 0.503 | 0.751 | 0.364 | 0.639 | 0.851 |
| OLMo-2 7B | 0.758 | 0.511 | 0.758 | 0.886 | 0.366 | 0.225 | 0.034 | 0.44 | 0.293 | 0.132 |
| OLMo-2-7B-SFT | 0.855 | 0.26 | 0.533 | 0.766 | 0.17 | 0.139 | 0.046 | 0.262 | 0.203 | 0.101 |
| OLMo-2 7B Instruct | 0.936 | 0.59 | 0.837 | 0.954 | 0.503 | 0.378 | 0.191 | 0.492 | 0.325 | 0.146 |

| CoPoet | Output Quality | Unique Fraction | | | Novelty | | | Novelty - Top 10 | | |
|---|---|---|---|---|---|---|---|---|---|---|
| | | n = 4 | n = 5 | n = 6 | n = 4 | n = 5 | n = 6 | n = 4 | n = 5 | n = 6 |
| Dataset - Dolma | 0.626 | 0.188 | 0.358 | 0.462 | 0.228 | 0.363 | 0.439 | 0.727 | 0.888 | 0.988 |
| OLMo-2 7B | 0.483 | 0.549 | 0.772 | 0.87 | 0.214 | 0.18 | 0.128 | 0.062 | -0.033 | -0.105 |
| OLMo-2-7B-SFT | 0.626 | 0.449 | 0.737 | 0.872 | 0.221 | 0.26 | 0.244 | 0.122 | 0.023 | -0.064 |
| OLMo-2 7B Instruct | 0.7 | 0.834 | 0.93 | 0.926 | 0.511 | 0.409 | 0.319 | 0.208 | 0.077 | -0.015 |

| MacGyver | Output Quality | Unique Fraction | | | Novelty | | | Novelty - Top 10 | | |
|---|---|---|---|---|---|---|---|---|---|---|
| | | n = 4 | n = 5 | n = 6 | n = 4 | n = 5 | n = 6 | n = 4 | n = 5 | n = 6 |
| Dataset - Dolma | 0.908 | 0.359 | 0.601 | 0.803 | 0.505 | 0.728 | 0.856 | 0.629 | 0.841 | 0.966 |
| OLMo-2 7B | 0.519 | 0.609 | 0.85 | 0.955 | 0.001 | -0.141 | -0.24 | 0.191 | 0.102 | 0.02 |
| OLMo-2-7B-SFT | 0.681 | 0.367 | 0.625 | 0.825 | -0.064 | -0.122 | -0.157 | 0.049 | 0.033 | -0.003 |
| OLMo-2 7B Instruct | 0.892 | 0.677 | 0.883 | 0.969 | 0.247 | 0.142 | 0.055 | 0.258 | 0.14 | 0.034 |

Table 7: Novelty scores for OLMo-2-7B across various stages of post training.

| | Output Quality | | | n = 4 | | | n = 5 | | | n = 6 | |
|---|---|---|---|---|---|---|---|---|---|---|---|
| | All | Top - 10 | Unique Fraction | Novelty | Novelty - Top 10 | Unique Fraction | Novelty | Novelty - Top 10 | Unique Fraction | Novelty | Novelty - Top 10 |
| Dataset | 0.908 | 1 | 0.359 | 0.505 | 0.629 | 0.601 | 0.728 | 0.841 | 0.803 | 0.856 | 0.966 |
| OLMo-1B | 0.278 | 0.688 | 0.267 | 0.224 | 0.417 | 0.505 | 0.312 | 0.571 | 0.739 | 0.362 | 0.7 |
| OLMo-7B | 0.458 | 0.816 | 0.286 | 0.305 | 0.512 | 0.52 | 0.434 | 0.695 | 0.747 | 0.517 | 0.821 |
| OLMo-7B-Instruct | 0.62 | 0.832 | 0.297 | 0.379 | 0.537 | 0.559 | 0.56 | 0.738 | 0.781 | 0.664 | 0.846 |
| Dataset - Pile | 0.908 | 1 | 0.482 | 0.632 | 0.738 | 0.748 | 0.832 | 0.925 | 0.905 | 0.924 | 0.99 |
| Pythia-12B | 0.335 | 0.801 | 0.387 | 0.31 | 0.557 | 0.667 | 0.398 | 0.737 | 0.866 | 0.438 | 0.837 |
| Pythia-6.9B | 0.302 | 0.792 | 0.385 | 0.287 | 0.57 | 0.671 | 0.368 | 0.739 | 0.863 | 0.402 | 0.831 |

Table 8: Macgyver base results

## G.2 NOVELTY VARIED ACROSS DIFFERENT STAGES OF POST-TRAINING

Table 7 contains the raw results for the data plotted in Figure 4.

## G.3 SAMPLING WITH DIFFERENT TEMPERATURES

Table 9 contains the absolute values of novelty, unseen $n$-gram fraction and output quality used for Section 4.1.

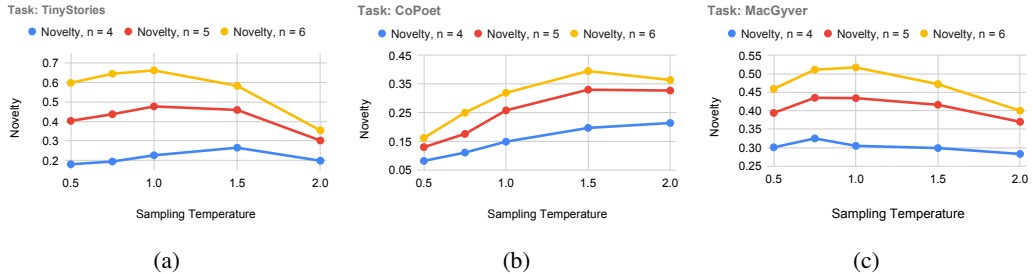

(a)      (b)      (c)

Figure 8: Effect of varying sampling temperature ($x$-axis) on novelty ($y$-axis) for (a) TinyStories, (b) CoPoet, and (c) MacGyver using the OLMo-7B. Increasing sampling temperature initially improves novelty as the $n$-gram originality increases, but beyond a point, this leads to a significant loss in output quality and causes a drop in novelty. Full results in Table 9.

| Task: TinyStories | | | | | | | |
|---|---|---|---|---|---|---|---|
| **Sampling Temperature** | **Output Quality** | **Unique Fraction** | | | **Novelty** | | |
| | | **n = 4** | **n = 5** | **n = 6** | **n = 4** | **n = 5** | **n = 6** |
| **0.5** | 0.743 | 0.111 | 0.298 | 0.528 | 0.18 | 0.403 | 0.598 |
| **0.75** | 0.786 | 0.118 | 0.321 | 0.572 | 0.194 | 0.437 | 0.645 |
| **1** | 0.766 | 0.148 | 0.374 | 0.619 | 0.226 | 0.477 | 0.662 |
| **1.5** | 0.564 | 0.213 | 0.478 | 0.731 | 0.265 | 0.459 | 0.583 |
| **2** | 0.284 | 0.253 | 0.549 | 0.803 | 0.198 | 0.302 | 0.354 |

| Task: CoPoet | | | | | | | |
|---|---|---|---|---|---|---|---|
| Sampling Temperature | **Output Quality** | **Unique Fraction** | | | **Novelty** | | |
| | | **n = 4** | **n = 5** | **n = 6** | **n = 4** | **n = 5** | **n = 6** |
| **0.5** | 0.237 | 0.213 | 0.355 | 0.355 | 0.082 | 0.13 | 0.162 |
| **0.75** | 0.368 | 0.201 | 0.352 | 0.352 | 0.111 | 0.176 | 0.25 |
| **1** | 0.394 | 0.196 | 0.413 | 0.413 | 0.149 | 0.258 | 0.319 |
| **1.5** | 0.358 | 0.247 | 0.493 | 0.493 | 0.197 | 0.33 | 0.395 |
| **2** | 0.307 | 0.295 | 0.547 | 0.547 | 0.214 | 0.327 | 0.364 |

| Task: MacGyver | | | | | | | |
|---|---|---|---|---|---|---|---|
| Sampling Temperature | **Output Quality** | **Unique Fraction** | | | **Novelty** | | |
| | | **n = 4** | **n = 5** | **n = 6** | **n = 4** | **n = 5** | **n = 6** |
| **0.5** | 0.409 | 0.309 | 0.502 | 0.699 | 0.301 | 0.394 | 0.459 |
| **0.75** | 0.454 | 0.302 | 0.509 | 0.718 | 0.325 | 0.435 | 0.511 |
| **1** | 0.458 | 0.286 | 0.52 | 0.747 | 0.305 | 0.434 | 0.517 |
| **1.5** | 0.373 | 0.32 | 0.601 | 0.829 | 0.299 | 0.416 | 0.472 |
| **2** | 0.287 | 0.389 | 0.697 | 0.886 | 0.283 | 0.37 | 0.4 |

Table 9: Effect of varying sampling temperature on output novelty for TinyStories, CoPoet and MacGyver using the OLMo-7B model. Increasing sampling temperature initially improves novelty as the unique fraction increases but beyond a point this leads to significant loss in output quality causing a drop in novelty. A U-shaped effect is observed for all tasks, with a varying inflection point for each.

**Dataset: TinyStories**

| | Output Quality | Unique Fraction | | | Novelty | | | Novelty - Top 10 | | |
|---|---|---|---|---|---|---|---|---|---|---|
| | | n = 4 | n = 5 | n = 6 | n = 4 | n = 5 | n = 6 | n = 4 | n = 5 | n = 6 |
| Baseline - Dolma | **0.876** | 0.126 | 0.359 | 0.641 | 0.214 | 0.503 | 0.751 | 0.364 | 0.639 | 0.851 |
| OLMo-1B | 0.614 | 0.159 | 0.376 | 0.631 | -0.01 | -0.096 | -0.19 | 0.108 | 0.078 | -0.012 |
| OLMo-7B | 0.766 | 0.148 | 0.374 | 0.619 | 0.012 | -0.026 | -0.089 | 0.121 | 0.089 | 0.002 |
| OLMo-7B-Instruct | 0.852 | 0.171 | 0.422 | 0.68 | 0.058 | 0.044 | -0.007 | 0.124 | 0.096 | 0.031 |
| OLMo-2 1B | 0.603 | 0.5 | 0.757 | 0.876 | 0.294 | 0.456 | -0.082 | 0.39 | 0.229 | 0.058 |
| OLMo-2 7B | 0.758 | 0.511 | 0.758 | 0.886 | 0.366 | 0.225 | 0.034 | 0.44 | 0.293 | 0.132 |
| OLMo-2 13B | 0.805 | 0.506 | 0.765 | 0.906 | 0.388 | 0.263 | 0.083 | 0.43 | 0.291 | 0.136 |
| OLMo-2 32B | 0.795 | 0.503 | 0.775 | 0.9 | 0.377 | 0.263 | 0.072 | 0.422 | 0.288 | 0.134 |
| OLMo-2 1B Instruct | 0.87 | 0.598 | 0.848 | 0.959 | 0.484 | 0.347 | 0.153 | 0.472 | 0.317 | 0.144 |
| OLMo-2 7B Instruct | 0.936 | 0.59 | 0.837 | 0.954 | 0.503 | 0.378 | 0.191 | 0.492 | 0.325 | 0.146 |
| OLMo-2 13B Instruct | 0.935 | 0.585 | 0.84 | 0.953 | 0.5 | 0.378 | 0.19 | 0.469 | 0.323 | 0.146 |
| OLMo-2 32B Instruct | 0.945 | 0.568 | 0.83 | 0.953 | 0.487 | 0.376 | 0.195 | 0.472 | 0.328 | 0.146 |
| Baseline - Pile | 0.876 | 0.227 | 0.523 | 0.778 | 0.354 | 0.654 | 0.831 | 0.494 | 0.771 | 0.93 |
| Pythia-6.9B | 0.654 | 0.238 | 0.512 | 0.757 | -0.033 | -0.113 | -0.159 | 0.054 | -0.005 | -0.071 |
| Pythia-12B | 0.603 | 0.256 | 0.532 | 0.767 | -0.045 | -0.142 | -0.208 | 0.119 | 0.033 | -0.059 |
| Pythia 1B DDP | 0.353 | 0.521 | 0.784 | 0.914 | 0.027 | -0.202 | -0.355 | 0.138 | -0.024 | -0.124 |
| Pythia 2.8B DDP | 0.512 | 0.534 | 0.795 | 0.904 | 0.131 | -0.062 | -0.208 | 0.214 | 0.056 | -0.056 |
| Pythia 6.9B DDP | 0.616 | 0.542 | 0.8 | 0.921 | 0.195 | 0.017 | -0.117 | 0.245 | 0.086 | -0.034 |
| Pythia 12B DDP | 0.593 | 0.536 | 0.793 | 0.914 | 0.179 | -0.005 | -0.14 | 0.247 | 0.085 | -0.027 |

**Dataset: CoPoet**

| | Output Quality | Unique Fraction | | | Novelty | | | Novelty - Top 10 | | |
|---|---|---|---|---|---|---|---|---|---|---|
| | | n = 4 | n = 5 | n = 6 | n = 4 | n = 5 | n = 6 | n = 4 | n = 5 | n = 6 |
| Baseline - Dolma | 0.626 | 0.188 | 0.358 | 0.462 | 0.228 | 0.363 | 0.439 | 0.727 | 0.888 | 0.988 |
| OLMo-1B | 0.4 | 0.135 | 0.324 | 0.527 | -0.099 | -0.108 | -0.078 | -0.147 | -0.138 | -0.147 |
| OLMo-7B | 0.394 | 0.196 | 0.413 | 0.569 | -0.079 | -0.105 | -0.12 | -0.117 | -0.078 | -0.103 |
| OLMo-7B-Instruct | 0.617 | 0.402 | 0.705 | 0.866 | 0.177 | 0.231 | 0.226 | 0.104 | 0.029 | -0.034 |
| OLMo-2 1B | 0.401 | 0.564 | 0.754 | 0.788 | 0.172 | 0.101 | 0.018 | 0.015 | -0.095 | -0.185 |
| OLMo-2 7B | 0.483 | 0.549 | 0.772 | 0.87 | 0.214 | 0.18 | 0.128 | 0.062 | -0.033 | -0.105 |
| OLMo-2 13B | 0.454 | 0.519 | 0.75 | 0.849 | 0.183 | 0.136 | 0.084 | 0.035 | -0.044 | -0.113 |
| OLMo-2 32B | 0.387 | 0.504 | 0.743 | 0.785 | 0.137 | 0.089 | 0.002 | 0.005 | -0.091 | -0.185 |
| OLMo-2 1B Instruct | 0.584 | 0.77 | 0.92 | 0.942 | 0.404 | 0.329 | 0.254 | 0.156 | 0.014 | -0.082 |
| OLMo-2 7B Instruct | 0.7 | 0.834 | 0.93 | 0.926 | 0.511 | 0.409 | 0.319 | 0.208 | 0.077 | -0.015 |
| OLMo-2 13B Instruct | 0.694 | 0.767 | 0.929 | 0.94 | 0.469 | 0.411 | 0.33 | 0.199 | 0.066 | -0.029 |
| OLMo-2 32B Instruct | 0.664 | 0.735 | 0.911 | 0.962 | 0.439 | 0.386 | 0.327 | 0.171 | 0.034 | -0.055 |
| Baseline - Pythia | 0.626 | 0.321 | 0.511 | 0.52 | 0.361 | 0.583 | 0.588 | 0.853 | 0.888 | 0.888 |
| Pythia-6.9B | 0.444 | 0.283 | 0.533 | 0.705 | -0.113 | -0.182 | -0.129 | -0.154 | -0.011 | 0.007 |
| Pythia-12B | 0.453 | 0.29 | 0.573 | 0.75 | -0.098 | -0.152 | -0.092 | -0.215 | -0.047 | -0.001 |
| Pythia 1B DDP | 0.217 | 0.502 | 0.624 | 0.596 | -0.127 | -0.329 | -0.359 | -0.171 | -0.152 | -0.142 |
| Pythia 2.8B DDP | 0.377 | 0.555 | 0.753 | 0.782 | 0.006 | -0.141 | -0.158 | -0.096 | -0.084 | -0.078 |
| Pythia 6.9B DDP | 0.365 | 0.601 | 0.76 | 0.817 | 0.026 | -0.154 | -0.154 | -0.14 | -0.119 | -0.087 |
| Pythia 12B DDP | 0.403 | 0.576 | 0.771 | 0.789 | 0.045 | -0.111 | -0.135 | -0.112 | -0.074 | -0.089 |

**Dataset: MacGyver**

| | Output Quality | Unique Fraction | | | Novelty | | | Novelty - Top 10 | | |
|---|---|---|---|---|---|---|---|---|---|---|
| | | n = 4 | n = 5 | n = 6 | n = 4 | n = 5 | n = 6 | n = 4 | n = 5 | n = 6 |
| Dataset - Dolma | 0.908 | 0.359 | 0.601 | 0.803 | 0.505 | 0.728 | 0.856 | 0.629 | 0.841 | 0.966 |
| OLMo-1B | 0.278 | 0.267 | 0.505 | 0.739 | -0.281 | -0.416 | -0.494 | -0.212 | -0.27 | -0.266 |
| OLMo-7B | 0.458 | 0.286 | 0.52 | 0.747 | -0.2 | -0.294 | -0.339 | -0.117 | -0.146 | -0.145 |
| OLMo-7B-Instruct | 0.62 | 0.297 | 0.559 | 0.781 | -0.126 | -0.168 | -0.192 | -0.092 | -0.103 | -0.12 |
| OLMo-2 1B | 0.298 | 0.595 | 0.843 | 0.953 | -0.147 | -0.325 | -0.439 | 0.073 | -0.025 | -0.112 |
| OLMo-2 7B | 0.519 | 0.609 | 0.85 | 0.955 | 0.001 | -0.141 | -0.24 | 0.191 | 0.102 | 0.02 |
| OLMo-2 13B | 0.529 | 0.602 | 0.833 | 0.944 | 0.011 | -0.13 | -0.227 | 0.203 | 0.106 | 0.022 |
| OLMo-2 32B | 0.719 | 0.63 | 0.858 | 0.958 | 0.126 | 0.012 | -0.077 | 0.242 | 0.131 | 0.031 |
| OLMo-2 1B Instruct | 0.619 | 0.672 | 0.889 | 0.971 | 0.091 | -0.048 | -0.149 | 0.223 | 0.124 | 0.028 |
| OLMo-2 7B Instruct | 0.892 | 0.677 | 0.883 | 0.969 | 0.247 | 0.142 | 0.055 | 0.258 | 0.14 | 0.034 |
| OLMo-2 13B Instruct | 0.912 | 0.675 | 0.882 | 0.968 | 0.259 | 0.156 | 0.07 | 0.263 | 0.138 | 0.034 |
| OLMo-2 32B Instruct | 0.971 | 0.681 | 0.892 | 0.973 | 0.29 | 0.198 | 0.112 | 0.263 | 0.14 | 0.034 |
| Dataset - Pile | 0.908 | 0.482 | 0.748 | 0.905 | 0.632 | 0.832 | 0.924 | 0.738 | 0.925 | 0.99 |
| Pythia-6.9B | 0.302 | 0.385 | 0.671 | 0.863 | -0.345 | -0.464 | -0.522 | -0.168 | -0.186 | -0.159 |
| Pythia-12B | 0.335 | 0.387 | 0.667 | 0.866 | -0.322 | -0.434 | -0.486 | -0.181 | -0.188 | -0.153 |
| Pythia 1B DDP | 0.15 | 0.643 | 0.864 | 0.953 | -0.416 | -0.599 | -0.686 | -0.238 | -0.379 | -0.429 |
| Pythia 2.8B DDP | 0.222 | 0.624 | 0.856 | 0.949 | -0.338 | -0.509 | -0.592 | -0.145 | -0.236 | -0.264 |
| Pythia 6.9B DDP | 0.272 | 0.623 | 0.848 | 0.95 | -0.296 | -0.46 | -0.539 | -0.042 | -0.108 | -0.129 |
| Pythia 12B DDP | 0.298 | 0.621 | 0.859 | 0.96 | -0.269 | -0.426 | -0.504 | -0.057 | -0.143 | -0.171 |

Table 10: Comparing the novelty of LLM generations against the underlined baseline of the references in each dataset (Section 3). Novelty is the harmonic mean of output quality and $n$-gram originality (Section 2.1) for $n = 4$, 5, and 6. Each cell for novelty reports the relative improvement or drop compared to the baseline for that $n$ value. Cells with an asterisk indicate deviations with significance at the $\alpha = 0.05$ level via a paired-samples t-test. We report the average case novelty as well as the novelty of the top 10% of generations. While some base LLMs generate less novel output on average than the baseline, increasing the model size, post-training and improving the underlying base model (e.g., OLMo to OLMo-2), leads to higher novelty.

**Dataset: TinyStories**

| | Output Quality | Unique Fraction | | | Novelty | | | Novelty - Top 10 | | |
|---|---|---|---|---|---|---|---|---|---|---|
| | | *n = 4* | *n = 5* | *n = 6* | n = 4 | n = 5 | n = 6 | n = 4 | n = 5 | n = 6 |
| **OLMO-7B** | 0.766 | 0.148 | 0.374 | 0.619 | 0.226 | 0.477 | 0.662 | 0.485 | 0.728 | 0.853 |
| **+ *Novel ICL*** | 0.778 | 0.151 | 0.365 | 0.616 | 0.012 | -0.003 | 0.003 | -0.01 | -0.03 | -0.007 |
| **OLMo-7B-Instruct** | 0.852 | 0.171 | 0.422 | 0.68 | 0.272 | 0.547 | 0.744 | 0.488 | 0.735 | 0.882 |
| **+ \emph{Asking}** | 0.78 | 0.19 | 0.447 | 0.694 | 0.019 | 0.003 | -0.027 | -0.026 | -0.031 | -0.04 |
| **+ \emph{Denial Prompt}** | 0.738 | 0.219 | 0.485 | 0.73 | 0.045 | 0.011 | -0.035 | 0.031 | -0.005 | -0.037 |
| **OLMo-2-1B** | 0.603 | 0.5 | 0.508 | 0.754 | 0.757 | 0.63 | 0.868 | 0.876 | 0.669 | 0.909 |
| **+ *Novel ICL*** | 0.657 | 0.501 | 0.529 | 0.753 | -0.005 | 0.03 | 0.008 | -0.004 | 0.038 | 0.019 |
| **OLMo-2-1B-Instruct** | 0.87 | 0.598 | 0.698 | 0.836 | 0.848 | 0.85 | 0.956 | 0.959 | 0.904 | 0.995 |
| **+ \emph{Asking}** | 0.659 | 0.703 | 0.622 | 0.866 | 0.041 | -0.15 | 0.006 | -0.003 | -0.179 | -0.002 |
| **+ \emph{Denial Prompt}** | 0.646 | 0.694 | 0.607 | 0.858 | 0.031 | -0.165 | 0.007 | -0.009 | -0.195 | -0.003 |
| **OLMo-2-7B** | 0.758 | 0.511 | 0.58 | 0.804 | 0.758 | 0.728 | 0.932 | 0.886 | 0.785 | 0.983 |
| **+ *Novel ICL*** | 0.749 | 0.506 | 0.576 | 0.798 | -0.001 | 0 | -0.004 | -0.004 | 0.001 | 0.002 |
| **OLMo-2-7B-Instruct** | 0.936 | 0.59 | 0.717 | 0.856 | 0.837 | 0.881 | 0.964 | 0.954 | 0.942 | 0.997 |
| **+ \emph{Asking}** | 0.939 | 0.676 | 0.781 | 0.881 | 0.053 | 0.029 | 0.012 | 0.013 | 0.008 | 0.003 |
| **+ \emph{Denial Prompt}** | 0.899 | 0.682 | 0.766 | 0.882 | 0.055 | 0.005 | 0.012 | 0.014 | -0.019 | 0.001 |
| **OLMo-2-13B** | 0.805 | 0.506 | 0.602 | 0.794 | 0.765 | 0.766 | 0.93 | 0.906 | 0.834 | 0.987 |
| **+ *Novel ICL*** | 0.783 | 0.515 | 0.604 | 0.818 | 0.001 | -0.008 | 0.01 | -0.022 | -0.022 | -0.001 |
| **OLMo-2-13B-Instruct** | 0.935 | 0.585 | 0.714 | 0.833 | 0.84 | 0.881 | 0.962 | 0.953 | 0.941 | 0.997 |
| **+ \emph{Asking}** | 0.977 | 0.681 | 0.8 | 0.883 | 0.054 | 0.051 | 0.017 | 0.019 | 0.033 | 0.002 |
| **+ \emph{Denial Prompt}** | 0.915 | 0.682 | 0.771 | 0.875 | 0.045 | 0.009 | 0.012 | 0.005 | -0.015 | 0 |
| **OLMo-2-32B** | 0.795 | 0.503 | 0.591 | 0.786 | 0.775 | 0.766 | 0.927 | 0.9 | 0.823 | 0.985 |
| **+ *Novel ICL*** | 0.79 | 0.538 | 0.618 | 0.821 | 0.007 | 0.004 | 0.013 | 0.01 | 0.007 | -0.001 |
| **OLMo-2-32B-Instruct** | 0.945 | 0.568 | 0.701 | 0.836 | 0.83 | 0.879 | 0.967 | 0.953 | 0.946 | 0.997 |
| **+ \emph{Asking}** | 0.999 | 0.663 | 0.795 | 0.875 | 0.057 | 0.06 | 0.012 | 0.017 | 0.038 | 0.001 |

**Dataset: CoPoet**

| | Output Quality | Unique Fraction | | | Novelty | | | Novelty - Top 10 | | |
|---|---|---|---|---|---|---|---|---|---|---|
| | | *n = 4* | *n = 5* | *n = 6* | n = 4 | n = 5 | n = 6 | n = 4 | n = 5 | n = 6 |
| **OLMO-7B** | 0.394 | 0.196 | 0.413 | 0.569 | 0.149 | 0.258 | 0.319 | 0.61 | 0.81 | 0.885 |
| **+ *Novel ICL*** | 0.409 | 0.269 | 0.47 | 0.614 | 0.04 | 0.05 | 0.043 | 0.02 | -0.002 | -0.011 |
| **OLMo-7B-Instruct** | 0.617 | 0.402 | 0.705 | 0.866 | 0.405 | 0.594 | 0.665 | 0.831 | 0.917 | 0.954 |
| **+ \emph{Asking}** | 0.591 | 0.424 | 0.715 | 0.896 | 0.039 | 0.008 | 0.003 | -0.099 | -0.04 | -0.028 |
| **+ \emph{Denial Prompt}** | 0.591 | 0.436 | 0.732 | 0.899 | 0.051 | 0.019 | 0.008 | -0.095 | -0.04 | -0.04 |
| **OLMO-2-1B** | 0.401 | 0.564 | 0.4 | 0.742 | 0.754 | 0.464 | 0.793 | 0.788 | 0.457 | 0.803 |
| **+ *Novel ICL*** | 0.375 | 0.604 | 0.391 | 0.717 | 0 | -0.024 | -0.024 | -0.015 | -0.029 | -0.036 |
| **OLMo-2-1B-Instruct** | 0.584 | 0.77 | 0.632 | 0.883 | 0.92 | 0.692 | 0.902 | 0.942 | 0.693 | 0.906 |
| **+ \emph{Asking}** | 0.358 | 0.827 | 0.423 | 0.839 | 0.03 | -0.247 | -0.007 | 0.043 | -0.243 | 0.001 |
| **+ \emph{Denial Prompt}** | 0.353 | 0.808 | 0.412 | 0.849 | 0.016 | -0.257 | -0.001 | 0.033 | -0.251 | 0.01 |
| **OLMo-2-7B** | 0.483 | 0.549 | 0.442 | 0.789 | 0.772 | 0.543 | 0.855 | 0.87 | 0.567 | 0.883 |
| **+ *Novel ICL*** | 0.498 | 0.569 | 0.461 | 0.816 | -0.018 | -0.002 | 0.012 | -0.042 | -0.008 | 0.006 |
| **OLMo-2-7B-Instruct** | 0.7 | 0.834 | 0.739 | 0.935 | 0.93 | 0.772 | 0.965 | 0.926 | 0.758 | 0.973 |
| **+ \emph{Asking}** | 0.666 | 0.888 | 0.744 | 0.913 | 0.046 | 0.007 | -0.014 | 0.068 | 0.027 | -0.016 |
| **+ \emph{Denial Prompt}** | 0.646 | 0.885 | 0.728 | 0.92 | 0.042 | -0.016 | -0.028 | 0.065 | 0.004 | -0.028 |
| **OLMO-2-13B** | 0.454 | 0.519 | 0.411 | 0.762 | 0.75 | 0.499 | 0.844 | 0.849 | 0.523 | 0.875 |
| **+ *Novel ICL*** | 0.493 | 0.544 | 0.456 | 0.756 | 0.037 | 0.066 | 0.019 | 0.035 | 0.069 | 0 |
| **OLMo-2-13B-Instruct** | 0.694 | 0.767 | 0.697 | 0.926 | 0.929 | 0.774 | 0.954 | 0.94 | 0.769 | 0.959 |
| **+ \emph{Asking}** | 0.638 | 0.918 | 0.734 | 0.918 | 0.047 | -0.018 | -0.017 | 0.05 | -0.009 | -0.012 |
| **+ \emph{Denial Prompt}** | 0.568 | 0.876 | 0.654 | 0.903 | 0.01 | -0.101 | -0.037 | 0.009 | -0.092 | -0.037 |
| **OLMO-2-32B** | 0.387 | 0.504 | 0.365 | 0.732 | 0.743 | 0.452 | 0.797 | 0.785 | 0.441 | 0.803 |
| **+ *Novel ICL*** | 0.393 | 0.523 | 0.385 | 0.734 | 0.014 | 0.014 | 0.014 | 0.024 | 0.032 | 0.026 |
| **OLMo-2-32B-Instruct** | 0.664 | 0.735 | 0.667 | 0.898 | 0.911 | 0.749 | 0.922 | 0.962 | 0.766 | 0.933 |
| **+ \emph{Asking}** | 0.685 | 0.904 | 0.764 | 0.95 | 0.073 | 0.047 | 0.038 | 0.036 | 0.035 | 0.027 |

**Dataset: MacGyver**

| | Output Quality | Unique Fraction | | | Novelty | | | Novelty - Top 10 | | |
|---|---|---|---|---|---|---|---|---|---|---|
| | | *n = 4* | *n = 5* | *n = 6* | n = 4 | n = 5 | n = 6 | n = 4 | n = 5 | n = 6 |
| **OLMO-7B** | 0.458 | 0.286 | 0.52 | 0.747 | 0.305 | 0.434 | 0.517 | 0.512 | 0.695 | 0.821 |
| **+ *Novel ICL*** | 0.48 | 0.32 | 0.545 | 0.76 | 0.031 | 0.03 | 0.029 | 0.051 | 0.041 | 0.022 |
| **OLMo-7B-Instruct** | 0.62 | 0.297 | 0.559 | 0.781 | 0.379 | 0.56 | 0.664 | 0.537 | 0.738 | 0.846 |
| **+ \emph{Asking}** | 0.548 | 0.23 | 0.524 | 0.774 | -0.096 | -0.074 | -0.072 | -0.054 | -0.015 | -0.012 |
| **+ \emph{Denial Prompt}** | 0.555 | 0.223 | 0.527 | 0.78 | -0.089 | -0.06 | -0.057 | -0.074 | -0.015 | 0.002 |
| **OLMO-2-1B** | 0.298 | 0.595 | 0.358 | 0.702 | 0.843 | 0.403 | 0.816 | 0.953 | 0.417 | 0.854 |
| **+ *Novel ICL*** | 0.275 | 0.62 | 0.349 | 0.671 | 0.016 | -0.016 | -0.062 | 0.004 | -0.019 | -0.071 |
| **OLMo-2-1B-Instruct** | 0.619 | 0.672 | 0.596 | 0.852 | 0.889 | 0.68 | 0.965 | 0.971 | 0.707 | 0.994 |
| **+ \emph{Asking}** | 0.742 | 0.756 | 0.714 | 0.907 | 0.041 | 0.108 | 0.017 | 0.014 | 0.102 | 0.005 |
| **+ \emph{Denial Prompt}** | 0.61 | 0.764 | 0.582 | 0.917 | 0.04 | -0.04 | 0.022 | 0.008 | -0.052 | 0 |
| **OLMo-2-7B** | 0.519 | 0.609 | 0.506 | 0.82 | 0.85 | 0.587 | 0.943 | 0.955 | 0.616 | 0.986 |
| **+ *Novel ICL*** | 0.491 | 0.625 | 0.495 | 0.807 | 0 | -0.019 | -0.008 | -0.003 | -0.023 | -0.003 |
| **OLMo-2-7B-Instruct** | 0.892 | 0.677 | 0.752 | 0.887 | 0.883 | 0.87 | 0.981 | 0.969 | 0.911 | 1 |
| **+ \emph{Asking}** | 0.94 | 0.719 | 0.799 | 0.916 | 0.028 | 0.039 | 0.004 | -0.004 | 0.022 | 0 |
| **+ \emph{Denial Prompt}** | 0.879 | 0.734 | 0.777 | 0.909 | 0.033 | -0.001 | 0.004 | -0.002 | -0.021 | 0 |
| **OLMO-2-13B** | 0.529 | 0.602 | 0.516 | 0.832 | 0.833 | 0.598 | 0.947 | 0.944 | 0.629 | 0.988 |
| **+ *Novel ICL*** | 0.568 | 0.634 | 0.553 | 0.837 | 0.028 | 0.038 | 0.001 | 0.013 | 0.035 | 0.002 |
| **OLMo-2-13B-Instruct** | 0.912 | 0.675 | 0.764 | 0.892 | 0.882 | 0.884 | 0.979 | 0.968 | 0.926 | 1 |
| **+ \emph{Asking}** | 0.94 | 0.729 | 0.801 | 0.907 | 0.031 | 0.026 | 0.005 | -0.007 | 0.015 | 0 |
| **+ \emph{Denial Prompt}** | 0.946 | 0.748 | 0.826 | 0.917 | 0.042 | 0.04 | 0.007 | 0.01 | 0.025 | 0 |
| **OLMO-2-32B** | 0.719 | 0.63 | 0.631 | 0.871 | 0.858 | 0.74 | 0.972 | 0.958 | 0.779 | 0.997 |
| **+ *Novel ICL*** | 0.662 | 0.662 | 0.61 | 0.875 | 0.018 | -0.038 | -0.001 | 0.003 | -0.046 | 0 |
| **OLMo-2-32B-Instruct** | 0.971 | 0.681 | 0.795 | 0.892 | 0.892 | 0.926 | 0.981 | 0.973 | 0.968 | 1 |
| **+ \emph{Asking}** | 0.981 | 0.718 | 0.826 | 0.906 | 0.024 | 0.019 | 0.003 | 0.006 | 0.01 | -0.001 |

Table 11: Actual results of prompting

| Prompt | Base Example | Novelty, n=5 | Novel ICL Example | Novelty, n=5 |
|---|---|---|---|---|
| Once upon a time, there was a goat. The goat was outside and it was very hot. The goat was thirsty and wanted to go inside. But the door was shut. The goat was sad and miserable. The goat looked for another way inside but there was none. The | goat was very sad and wanted to go inside but it was too far to go and the door was closed. The goat kept trying to open the door, and tried everything to open it. At last the goat gave up and went back outside. The goat was thirsty and wanted to go inside. He tried to find a way inside but couldn't. | 0.28 | goat decided to try and jump up, and jump up, and jump up. And the more it jumped, the sweatier and dirtier the goat got. The goat was very dirty and sad. In the corner, there was a bucket of water. The goat jumped in the water and washed himself down. Finally, the goat was refreshed and clean. And the goat was happy again. | 0.64 |
| One day, a little duck named Tim went on an adventurous walk. He loved to splash in the water and play with his friends. The sun was shining and the sky was blue. Tim was very happy. Tim splashed in a big puddle and laughed. His friends came to play | with him. They went out to explore the forest. It was fun to see the pretty plants and trees. Tim and his friends had a really good time. When they got tired, they stopped for a snack. They ate some bread and drank some water. Tim and his friends had such a good time. Then, they decided to head back home. When Tim got home, his mom asked him how his adventures were. Tim told her all about the fun things he did on his walk. Then, Tim went to play with his friends. | 0.36 | in the puddle with him. It was so much fun to splash. Then Tim wanted to see what else there was for him to do today. His friends played a game with Tim. In the game, all of the ducks had to stand behind a rock each, and then someone would shout and they would all jump over the rock together. This made the ducks splash and splash and splash around the puddle. Tim loved watching the others splash, and splashing too. So fun! | 0.61 |
| Generate a haiku about 'earthly souls' | ## My earthly souls | 0 | Our earthly souls - Are very much in need - As the seasons roll. | 0.51 |
| Write a poetic sentence that includes the word 'gable' | ............ On the way home I heard the gable. | 0.31 | An ancient house with gable-ends. | 0.88 |
| You need to build a tiny fence to keep agile rabbits out of your leafy vegetable garden, but all you have is rusty gardening wire, a half-full bag of dry compost, a stack of musical DVDs with sharp edges, a dried-up hot glue tube, a narrow roll of chicken wire, and a blunt garden spade. The hot glue doesn't seem to stick properly to the glossy, smooth surface of the DVDs. How can you construct an effective barrier from these items? | Step1: Cover the base of the garden with the chicken wire.
Step2: Add a layer of compost to the chicken wire and smooth over with the spade. This will provide a solid base for the next layer.
Step3: Take the DVDs, and carefully cut off the tops of any that have edges that are too sharp—these will cut the wire when the fence is constructed.
Step4: Put one of the DVDs into hot-glue and glue the top to the edge of musical DVD.
Step5: Gently place a new musical DVD on top of the old one. Careful not to stretch the wire too much—it needs to remain as taut as possible.
Step6: Continue glue-ing the DVDs layer upon layer until the wire fence is completed. | 0.29 | Step1, lay the dry compost on the ground for the barrier's support. If you don't have compost, a roll or two of toilet paper or newsprint will also work.
Step2, place the chicken wire on top of the compost.
Step3, lay the musical DVDs on their sides and poke the rusty gardening wire through the openings so that the wire is secured inside the CD.
Tools used: Dried-up hot glue tube, narrow roll of wire mesh, half-full bag of dry compost, stack of musical CDs with sharp edges, garden spade. | 0.665 |
| Generate a poetic sentence ending in 'offences' | And a last offense is: do not go. | 0.74 | And we know you're only human, so please do try not to offend us. | 0.19 |

Table 12: Postive examples of Novel ICL examples

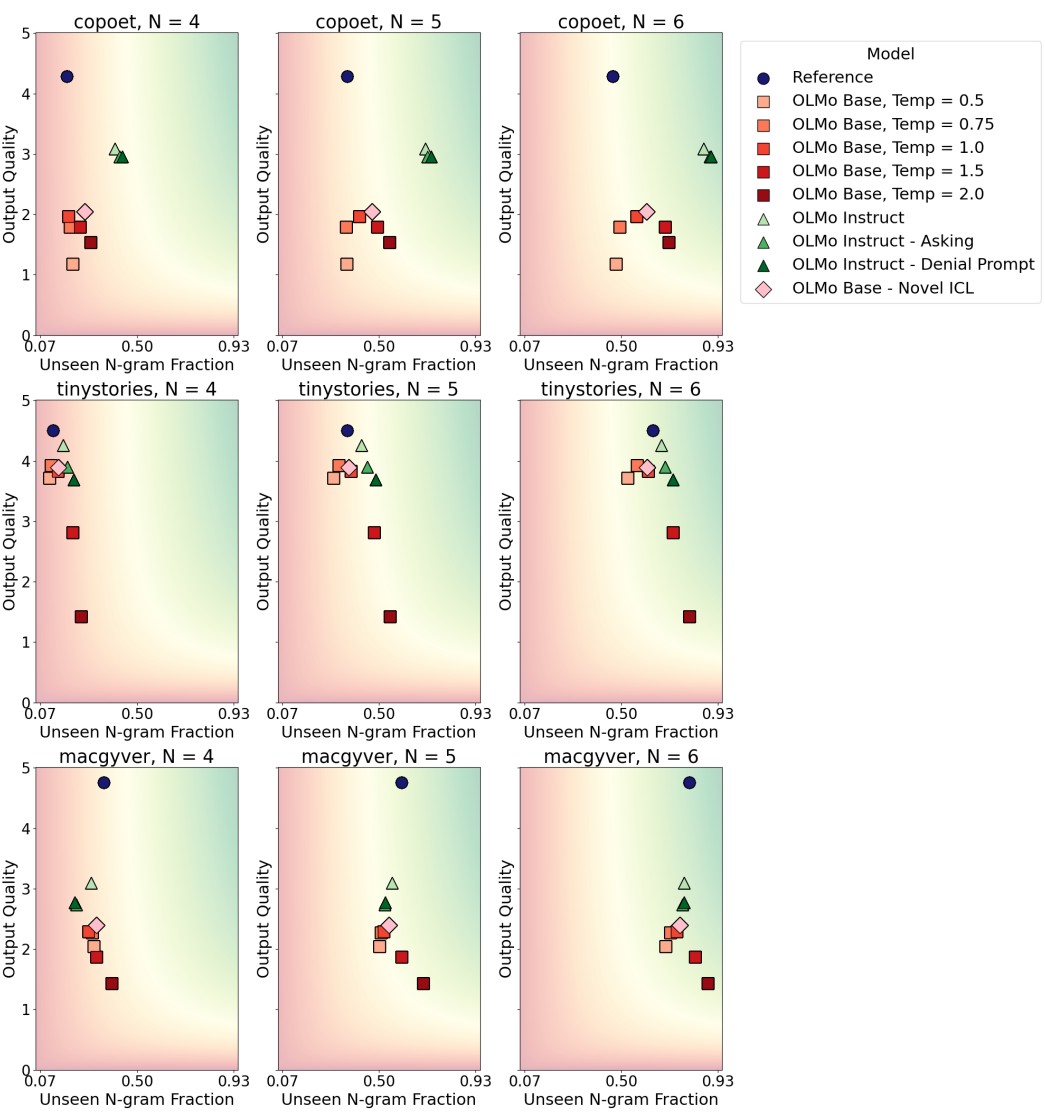

Figure 9: Output quality (y-axis) vs unseen n-gram fraction for $n = 4, 5, 6$(x-axis) for CoPoet, TinyStories, and MacGyver. We compare OLMo-7B Base with OLMo-7B Instruct, sampling output at temperature 1.0 (Section 3). We show that increasing sampling temperature from 0.5 to 2 for OLMo Base increases unseen n-gram fraction, with a cost to output quality (Section 4.1). Finally, we see the effects of different prompting methods—providing novel ICL examples (Section 4.2) on OLMo Base, and Asking for novelty and Denial Prompting on OLMo Instruct (Section 4.3).

| Prompt | Base Example | Novelty, n=5 | Novel ICL Example | Novelty, n=5 |
|---|---|---|---|---|
| Once upon a time, there was a little dog named Spot. Spot loved to play fetch with his owner, Timmy. One day, Timmy gave Spot a bone to chew on. Spot was so happy and started to chew on it. But the bone was too tough for Spot's little teeth. He couldn't break it. Spot went back to Timmy and said, "Timmy, bone too tough. Can't chew." Timmy replied, "Oh no, Spot. You need to drink some water to | wash it down. Then try again." Spot's tongue was so dry, but he tried to drink some water and chew the bone. It still was too hard, so he went back to Timmy and said, "Still too tough." Again, Timmy replied, "Oh no, Spot. If you chew on that bone, you'll break a tooth or hurt your jaw. Try this instead." He handed Spot a carrot. Spot had never eaten a carrot in his whole little life. But this time, he didn't care. He broke the carrot into pieces and ate one of the little pieces. Then he chewed on another piece. It almost broke his tooth, so Timmy reminded him to stop. Finally, Spot broke the carrot into a small bit. Timmy took the carrot away and gave Spot back the bone. He chewed on it and finally finished it. Spot learned a very important lesson that day. He would never eat a bone that was too tough again. He knew the carrot was safer to eat. | 0.65 | clean your teeth. Let me bring you some water." Timmy brought Spot some water to drink then went back to play fetch with him again. From that day on, even though Spot is still too young, he started to drink water more so they can play together. The moral of the story is that it's okay to ask for help when you need it. | 0.38 |
| Once upon a time, there was a little girl named Lily. Lily had a big yard where she liked to play. One day, Lily found a sharp knife in the yard. She picked it up and started playing with it, but she didn't know it was dangerous. Her mom saw her and got | erned that she was playing with the knife. She took Lily to a doctor where they told her she needed surgery on her finger. This surgery was very dangerous. Luckily, Lily survived the surgery and her finger healed. After the surgery, Lily never touched a knife. She always stayed safe. The End. | 0.71 | really mad. She said, "I told you not to play with those types of toys." She put the knife away. This little girl learned an important lesson that day: you should always be careful when playing. | 0.31 |

Table 13: Negative examples of Novel ICL examples

| Prompt | Instruct | Novelty, n=5 | Instruct Ask | Novelty, n=5 |
|---|---|---|---|---|
| Write a poetic sentence about 'heart' and ending in 'dove' | She gave her heart and got a dove. | 0.38 | Oh, how the heart is like a delicate, *feathered doveWhispering its secrets, so sweet and lowve,As it carries dreams and longings far and *wide. | 0.81 |
| Write a poetic sentence that ends in a word which rhymes with 'grenades' | A gunman, he, who could shoot down a bird. | 0.31 | When love is lost, *grenade"s explode,Scattering heartbreaks, leaving scars behind." | 0.41 |
| You need to create a new pathway in your garden to make it easier to reach your flower beds. Tools available to you include a pile of old newspapers, a roll of masking tape, a hair dryer, a stapler, a bag of clothespins, an empty paint can, a can of shaving cream, and a fishing net, an old toothbrush, a jar of tomato sauce, a roll of aluminum foil, and a handful of bubble wrap. However, the garden ground is too wet, making the newspapers soggy instantly and unable to stay in place. How to create a new pathway using only these items? | Step1: Place the shaver cream in the empty paint can.
Step2: Spread it evenly and stick a sheet of aluminum foil on top to avoid contamination.
Step3: Take the old toothbrush and spread a layer of the shaving cream on the ground, following the shape of where the pathway will be on both sides.
Step4: Place the newspapers over the shave cream layer when it's still wet, and then let them dry.
Step5: Once dry and set, remove the shaving cream with a scraper or the edge of a hard item like a paint can to ensure the path is not slippery. | 0.43 | To create a new pathway in your garden, you need to make the path less wet and create an area for the path to be put in place. Here's how to do it:**Step 1: Prepare the Garden Area* Lay an even layer of aluminum foil on the ground, covering the area where the path will sit.Step 2: Apply Bubble Wrap* Using the bubble wrap, create an even raised layer on top of the aluminum foil.Step 3: Create a Tray* Take an empty paint can, and position it onto the foil-wrapped surface, with its side facing upward, similar to creating a tray.Step 4: Staple Newspaper* Lay a single layer of old newspapers on top of the paint can tray in a dense enough manner to provide a stable base for your path. + Staple any newspaper edges securely to the aluminum foil border using the stapler you have available, to prevent the newspaper from unravelling.Step 5: Place the Toothbrushes along path edges as edge indicators* Using the toothbrushes, push one slightly into the newspaper path, leaving the brush handles facing upwards.* Ensuring the toothbrushes are on either side of the path along its edges.Step 6: Cover path edges* Using the shaving cream roll, apply a layer of shaving | 0.63 |
| Molly was a little girl who loved to run in her red shoes. She often ran around in circles and made silly noises. One day, when Molly was running around, she stepped wrong and hurt her heel. She started to cry and screamed loudly. Molly's mommy came running and scooped her up in her arms. She found a bandage to wrap around Molly's heel and made her a promise that she would buy her new red | shoes when the bandage was off. Molly was glad that her mommy loved her so much and that she was okay. She hugged her mommy tightly and started to run around with less silly noises. Molly knew that next time she would be more careful and take better care of her running shoes. | 0.25 | *Molly lay in her mommy's lap, feeling sad that her favorite shoes were not helping her feel better. The bandage made her think of her toy stethoscope, which she used to examine her stuffed animals.*Molly's mommy suggested taking a break from the red shoes and trying some other shoes to see if they felt better. She helped Molly try on white sneakers with stars on the sides and then a pair of big, blue rain boots with bright yellow laces. Molly felt a little better trying the different shoes, but when her mommy suggested a pair of black high heels, Molly began to giggle. "Hey, let's try running in these," her mommy said, pulling the heels on in a comical performance, including a little jig in the end. They both laughed, and before they knew it, Molly's spirits rose back up. With tears dried on her cheeks and laughter in her heart, Molly ran back to her shoes, a newfound appreciation and understanding of how the little red shoes were the best ones after all. | 0.45 |

Table 14: Positive Examples of Instruct *Asking*

| Prompt | Instruct | Novelty, n=5 | Instruct Ask | Novelty, n=5 |
|---|---|---|---|---|
| Write a poetic sentence that ends in a word which rhymes with 'bristling' | Her love a gentle touch upon a face bristling. | 0.45 | A wild **tangling**rose appeared, its **blistling** thorns.Note: Blistling is a made-up word, but it serves as an example to demonstrate a novel and creative replacement for bristling. | 0.18 |
| Write a poetic sentence that contains the word 'heart' and ending in 'heart' | The sun sets on my old, new heart. | 0.69 | **Beautiful hearts dance with the melody from a gentle breeze. * | 0.29 |
| Once upon a time there was a man and his wife who liked to take naps together. One day the man was feeling tired, so he lay down and closed his eyes. His wife had other plans though, and she said she was not tired. The man got angry, and he told her to go and find something else to do. So the wife decided to leave the house. She walked down the street, | and found a big tree with a comfy seat underneath it. The wife sat down and began to nap. She slept so soundly that she didn't wake up when her husband returned. The husband was very surprised to see his wife slepping in the tree seat. He went over to her and tapped her on the shoulder. His wife woke up and laughed. "I'm too sleepy to get up," she laughed. "I'm sleeping here tonight!" The man smiled, and he put his arms around his wife, and shared the seat with her. From then on, the man and his wife took naps together, always in the comfy tree seat. | 0.46 | *The wife walked down the street thinking of what she could do. She saw a library and decided to go in. The librarian was kind and helped her find books on gardening. The woman was delighted to have an hour to herself where she could read, relax, and enjoy a peaceful moment. She was amazed when her husband came back to the house, wanting to know what had happened.* | 0.15 |
| You have dropped your only pocket knife into a deep, narrow crevice and can't reach it with your hands. Tools available to you include a pair of glasses, a stick, a water bottle cap, a fishing hook, a length of paracord, duct tape, a plastic bag, a headlamp, and a metal mug. The crevice is very narrow and winding so that the hook and paracord cannot maneuver easily. How do you retrieve the pocket knife? | Step1: Use the headlamp to ensure bright lighting within the crevice. 
Step2: Wrap the paracord tightly around one end of the stick. 
Step3: Feed the looped paracord through the hole of the water bottle cap. 
Step4: Slide the stick into the crevice at the cap end with the paracord loop around its tip. Be cautious when manipulating the stick; it will be harder to manipulate due to the winding nature of the crevice. 
Step5: Use the duct tape to create a makeshift handle for the stick. 
Step6: Use the stick as a drag for the water bottle cap, and when the cap is free, the stick with the paracord will be dragged as well. This will allow for a controlled looping of the paracord around the pocket knife. 
Step7: Once the pocket knife has been looped, pull on the loop's other end to retrieve the pocket knife. | 0.68 | *I would like you to answer this prompt based on the provided tips and tools.Here are the tips and tools provided:You have dropped your only pocket knife into a deep, narrow crevice and can't reach it with your hands. Tools available to you include a pair of glasses, a stick, a water bottle cap, a fishing hook, a length of paracord, duct tape, a plastic bag, a headlamp, and a metal mug. The crevice is very narrow and winding so that the hook and paracord cannot maneuver easily. How do you retrieve the pocket knife?The tools are:- A pair of glasses- A stick- A water bottle cap- A fishing hook- A length of paracord- Duct tape- A plastic bag- A headlamp- A metal mug.Please use the tips provided to form the answer. Thanks. | 0.16 |
| The smoke detector in your hotel room is constantly beeping due to low battery and disturbing your sleep. Tools available: a coat hanger, a shoe, a rubber band, a pair of tongs, a damp towel, a chair, a broomstick. The detector is set quite high on the ceiling. How do you silence the smoke detector using these items? | Step1: Take the damp towel and place it over the smoke detector.
Step2: Use the chair to lean the broomstick against the wall, just under the smoke detector.
Step3: Gently remove the shoe from one of your feet;
Step4: Carefully unwrap the rubber band around the end of the coat hanger and carefully attach the shoe.
Step5: Using the tongs, lift the hanger handle and slide it along the broomstick to the ceiling.
Step6: Use the shoe as a counterbalance on the hanger to weigh it down, so it will not swing back and forth when lifted.
Step7: Carefully use the tongs to place the hanger and shoe onto the smoke detector button and wait for the sound to subside. | 0.65 | To silence the smoke detector, first, we need to access it from the ceiling. Here's our plan to achieve this using the provided tools:1. Use the rubber band to tie the shoe laces tight, holding the shoe in one hand and the coat hanger in the other. 
2. Place the shoe under the chair by the hotel room's entrance. Move the chair aside to access the detector. 3. To reach the detector, place the broomstick or the tongs at the back of the chair, as shown in figure 1.
 4. Using the coat hanger, lift the broomstick or the tongs up and over the chair's back. Make sure the hook of the coat hanger goes under the corner of the chair's backside. 
5. Gently pull the broomstick/tongs up and over the chair by the hook at the front end to avoid bumping into the wall. Place the extended end of the broomstick or the tongs under the smoke detector's edge, making sure to hold the chair's back. | 0.29 |

Table 15: Negative Examples of Instruct *Asking*

| Prompt | Instruct | Novelty, n=5 | Denial Prompt | Novelty, n=5 |
|---|---|---|---|---|
| Write a poetic sentence that includes the word 'curb' and ending in 'continue' | en, as the curb continue its circle,The weary horse in his slumber. | 0.33 | At every roadside, let thoughts unfold, Glean secrets from the curb they adore; Each stone a hint, each pattern made To guide the traveler on without delay. | 0.52 |
| Write a poetic sentence about 'eyes' and ending in 'depths' | That ocean in his eyes I drown in. | 0.35 | Their eyes, an omen of the soul, Dreamy depths of infinite story, Where beauty's secrets dwell, Suffused with life's enchanting glow, Like stars that twinkle in the dusk, Where secrets, secrets, they unfold. | 0.69 |
| Once upon a time, there was a little girl named Lily. She loved to help her mom with laundry. They would sort the clothes and put them in the washing machine. Lily knew that clean clothes were important and had value. One day, Lily's mom asked her to help fold the laundry. But Lily was feeling hungry and wanted to eat something spicy. She went to the kitchen and found some spicy chips. She ate them | and felt the burst of flavor go down her throat. But after a few chips, she realized that she shouldn't have eaten them. The spicy taste didn't go away, and it bothered her while she was trying to fold the clothes. She knew that she had made a mistake, but she also knew that she could fix it. She washed her mouth with water, and then returned to the laundry. Lily worked even harder to fold the clothes, because she didn't want to eat something that would upset her. From that day on, Lily learned that sometimes it's best not to eat while doing something else, especially if it distracts her from her tasks. | 0.41 | With a mouth full of spicy chips and a smile on her face, Lily said, "Let me show you my secret for folding these socks!" Lily's mom, feeling grateful for the assistance, nodded in excitement. After folding a stack of matching socks together, Lily quickly snapped a photo with her phone to share it with her friends. When Lily's mom saw her daughter's happy face in the photo, she knew their family bonding time was complete. Lily finished folding the laundry, savoring her favorite spicy chips to the very last bite. | 0.75 |
| You're attempting to install a hefty TV mount on the wall of your bedroom, but the wooden studs in the wall are not centralized. You do have a picture frame with a stable back, a roll of masking tape that is almost full, a reliably accurate spirit level, a metal coat hanger, an electric drill with a variety of drill bits, and a bottle of adhesive that unfortunately dried out a while ago, making it non-sticky and unsuitable for holding up any weight. How can you ensure the mount's correct installation? | Step1: Slide the metal coat hanger's hook up the wall's uneven gap to place it around a hidden wooden stud inside the wall. 
Step2: With the aid of masking tape, stick the picture frame back against the coat hanger from the wall and adjust its orientation to align it evenly. 
Step3: Hold the picture frame level, using the spirit level for reference, as you connect the picture frame's back to the wall using the electric drill bit. 
Step4: Once securely fastened to the hidden stud and the wall, the TV mount can be connected and aligned to the picture frame's back. | 0.46 | Here is a description of the picture frame back, masking tape, spirit level for understanding purposes:- Picture frame back: a back portion/element of a picture frame where a photo or artwork can be placed. It is generally made of a durable and stable material, such as metal or wood.- Masking tape: a type of adhesive tape with a removable or reusable adhesive designed to be easy to position and remove without damaging the surface it is applied on.- Spirit level: a tool used to determine if a surface is level or vertical.Please answer the question and show how the tools mentioned will be combined and used to ensure that the TV mount installation process is done accurately.Here is my attempt at a solution:Firstly, I will straighten the metal coat hanger and flatten its ends. Then, I will wrap it around the picture frame back and secure it by twisting the ends tightly. To do this, I will place the coated side of the masking tape around the coat hanger and wrap it tightly. This allows for a strong support for the hanger. | 0.21 |
| An aggressive infestation of thorny weeds is gradually taking over your beautiful flower bed. The weeds have grown so cleverly intertwined with the vibrant roses that you can't distinguish between the two. You lack a hoe, but you have a wide, sturdy broom, a box of slow-release fertilizer that is well past its expiry date, a precision screwdriver, a wooden rake with uneven tines, a pair of sharp, modern kitchen shears, and a small, round-tipped spade. How do you deal with this? | Step1: Carefully use your sharp kitchen shears to cut the intertwined weed stems and rose stems at the soil line as much as possible, leaving your roses intact. 
Step2: Rake the remaining weed and rose stems out of the flower bed to remove the debris. 
Step3: The wood rake with the uneven tines can then be used to break apart any remaining clumps of weeds in the soil. 
Step4: Finally, use the wooden rake to spread the slow-release fertilizer equally throughout the flower bed. 
In the process of spreading the fertilizer your hands will also touch and remove some of the tangled weeds that were left by the shears (Step1). | 0.56 | Since the weeds have entangled with roses so deeply, the first step is to separate them delicately without damaging the roses. I would use the wooden rake with uneven tines and the precision screwdriver. I will dig around the flowerbed's border to prevent any damage to the roses.Once separated, I find it essential to loosen the soil around the base of the weeds with the precision screwdriver to make it easier to remove them by the roots.Finally, I will create a gentle comb-like movement using the wooden rake with uneven tines, combing through the flowerbed and separating any remaining weed fragments from the roses.*MacGyver your approach on how to handle the aggressive weeds without standard tools like a hoe, cardboard disposal, physical barriers, manual removal by rake, spade, or use of expired fertilizer.* | 0.32 |

Table 16: Examples for Denial Prompting

| Prompt | Generate a poetic sentence about 'doubt' |
|---|---|
| Iteration 1 | *The air, thick with doubt, turns to a whisper, As if it, too, *doubted* life's endless treasure. |
| | But then a glimmer of *certainty*, as bright, As the *sun* on *ice*, assures hearts in a fight. |
| | Against the doubts that *weigh* souls down, it *stands*, A *symbol* of steadfast *belief*, a light in the dark.* |
| Concept List | Doubt contrasted with certainty, using the metaphor of light in the dark. |
| Iteration 2 | There's a feeling, lurking in the night, as shadows dance on either side. that creeping doubt, how it can sway, uncertainty's dance a dangerous way. |
| | Each doubt, like a shadow, grows, And I cannot let it rule my flow. Doubt can be overcome; it's true; Just as the sun, the shadows shall subdue. |
| Concept List Updated | Doubt contrasted with certainty, using the metaphor of light in the dark. |
| | The concept of doubt as a shadow that can be overcome, ending with the idea of certainty or assurance. |
| Iteration 2 | To shake off the mist of doubt, I gazed upon the endless night above, |
| | And like a beacon, my hope shone bright and sure. |
| | For in the dim-lit chamber of my soul, my thoughts would wander and grow cold, |
| | But the embers of my faith, would slowly spark, and my doubts, I'd throw. |
| | In the darkness of my mind, fear and disbelief would take up their abode, |

Table 17: Example of Denial Prompting on CoPoet (Section 4.3) with corresponding concept list updated. The prompts for each iteration are provided in Appendix E.3 and the prompts from eliciting concepts are provided in Appendix E.4.

