# OpenReview forum: "Measuring LLM Novelty As The Frontier Of Original And High-Quality Output"
_ICLR.cc/2026/Conference — ICLR 2026 Poster_

### Official Review · Reviewer_KYVM · 2025-10-26

**Soundness:** 3
**Presentation:** 3
**Contribution:** 2
**Rating:** 6
**Confidence:** 3

**Summary:**

This paper proposes a new metric to measure the novelty and originality of content (across different tasks) generated by LLMs.
Existing n-gram-based methods for originality may assign high scores to low-quality text, as they only consider whether the generated text appears in the training data or not.
To address this limitation, the paper introduces a harmonic mean between the n-gram-based originality metric and an LLM-based quality metric.
The authors evaluate their proposed metric on various models and show that increasing model size leads to improved performance in novelty.

**Strengths:**

The paper is easy to read.
The problem being addressed is interesting, as it is necessary to measure the quality of the content generated by the models

**Weaknesses:**

I did not find much clarity and usefulness in the proposed metric, and it is also unclear why the authors chose to use the harmonic mean.

Moreover, using an n-gram model to capture originality may not be an appropriate choice, since a model could generate a sentence that conveys the same meaning or context as one found in the training data, but with a different ordering of words.

**Questions:**

The authors use an LLM to quantify the quality of the generated text. However, is it possible for an LLM to assess both novelty and quality simultaneously?

I believe that an n-gram model may not effectively capture the context of a statement. Since a coherent and meaningful sentence can be expressed in many different ways, an n-gram–based approach might classify a statement as novel even when its underlying context is similar to one already present in the training data.

Could the authors also clarify why they chose to use the harmonic mean to combine the two metrics?

Here, the quality of the generated text is evaluated by another LLM model? How do you make sure, the assessment by the LLM model is accurate?

---

> ### Author Response · Authors · 2025-11-21
> **Response to Reviewer KYVM (1/1)**
>
> We appreciate your feedback and recognition of the strengths of our work. Here we discuss how we incorporated the same in our draft and respond to your concerns:
> - **(Weakness 1) On the utility of such a measure:** We believe that our measure of novelty is most helpful in providing a data-centric evaluation of capabilities as we continue to develop stronger models. Such a measure can also be used as a reward for training models in domains like creativity and scientific discovery, where it is desirable for the model to extrapolate from the training data to new insights.
> - **(Weakness 1, Question 3) On the choice of harmonic mean:** The choice of aggregation measure among harmonic, arithmetic and geometric mean was essentially arbitrary, with preference given to the harmonic mean as we intended to penalize generations where either one value (originality, or quality) was low.
> - **(Weakness 2, Question 2) On semantic measures of originality/limitations of n-grams:** We choose n-gram-based analysis because it is scalable to the size of pre- and post-training corpora. We acknowledge that n-grams are not perfect and that paraphrasing could change the distribution of occurrences; however, in expectation over the whole dataset, we hope these effects would average out and provide us with reliable insights comparing different models. While semantic measures of originality would work well, we are not aware of any methods that could scale to the size of contemporary model training datasets.  We detail this as a limitation in Appendix B. We also provide experiments using perplexity for originality in Appendix G.1 that show that our conclusions generalize to this measure as well. We are open to expanding our analysis further if there is a specific measure you could point us to.
> - **(Question 1) On using LLMs to gauge novelty:** The main reason not to use an LLM to gauge the originality axis was based on findings from prior work [1] that contemporary models, even from frontier labs, do not accurately match expert judgments of originality. While we train LLMs to match human quality judgments, originality is not traditionally a rubric that is optimized for in current preference tuning objectives so this is unlikely to improve naturally as we develop better models. Hence, we use the LLM for judging quality and exact indexing of the model training data for originality.
> - **(Question 4) On ensuring that judgments of quality are accurate:** We rigorously test that the LLM-as-judge evaluation matches human judgments in Appendix D. Through a user study with 11 annotators providing judgments on 100 examples of each task, we find that o3-mini obtains the highest correlation with aggregated human scores (when compared with various models from frontier labs) and use the same for our experiments. Finally, we find that frontier models are increasingly better at modeling user judgments (o1 < o3, Claude 3.7 < Claude 4), so we expect that this axis is more aligned with contemporary LLM progress.
>
> [1] Chakrabarty, Tuhin, et al. "Art or artifice? large language models and the false promise of creativity." Proceedings of the 2024 CHI Conference on Human Factors in Computing Systems. 2024.
>
> **Please let us know if you have any other questions that we can address. If not, we sincerely hope you can consider increasing your rating to 8 (Accept), as we believe that our contribution is significant to the open-science LLM community and LLMs X creativity communities.**

---

> > ### Comment · Reviewer_KYVM · 2025-11-26
> > **Thank you for the clarification**
> >
> > I want to thank the authors for their response; however, I am not convinced by the proposed method for evaluation. For this reason, I will stick with my original score. But I don't mind if the AC/PC are leaning toward the acceptance of the paper.

---

> > > ### Author Response · Authors · 2025-12-01
> > > **About the utilty of our metric**
> > >
> > > Thank you for responding to our rebuttal! To provide more evidence for the value of our evaluation, we also note an interesting case study. In [1], Section 2.3, the authors initially thought that GPT-5 had come up with a valid and novel proof which was later found to be copied from the training data, but unknown to the authors of the work. The quote from the paper, "We were very excited to have an exact solution to a question which we had been curious about for years, especially because one direction of the proof was due to an AI. However, we soon learned that the same tight bound (as well as a different proof of the matching upper bound) had appeared on Arxiv nearly 3 years previously in the short paper [Alo24]. .... Our experience illustrates a pitfall in using AI: although GPT-5 possesses enormous internal knowledge and the capability to locate even more using the internet, it may not always report the original information sources accurately. This has the potential to deceive even seasoned researchers into thinking their findings are novel. We expect that our experience is not unique, and urge others to take special care in attribution when working with LLM-assisted proofs." **We think this very directly motivates evaluation methods such as ours and hope that this assuages some of your concerns about why our metric could be very valuable when foundation models are trained on internet-scale data.**
> > >
> > > [1] "Early science acceleration experiments with GPT-5", https://cdn.openai.com/pdf/4a25f921-e4e0-479a-9b38-5367b47e8fd0/early-science-acceleration-experiments-with-gpt-5.pdf

---

### Official Review · Reviewer_ePsk · 2025-10-28

**Soundness:** 3
**Presentation:** 4
**Contribution:** 3
**Rating:** 8
**Confidence:** 5

**Summary:**

This paper introduces a new metric for evaluating the novelty of LLM generations by combining originality and quality. Originality is defined as the fraction of n-grams in the generated text that are not present in the model's training data (harmonic mean). Quality is measured via a task-specific, LLM-as-a-judge score. The proposed novelty metric is the harmonic mean of these two components. Using this framework, the authors conduct a thorough empirical study on open-data models (OLMo, OLMo-2, Pythia) across three creative tasks. Their findings indicate that model scale, improved base models, and post-training all contribute to higher novelty, primarily by improving output quality. Conversely, inference-time techniques like adjusting temperature or using novel in-context examples offer limited gains, often trading originality for quality.

**Strengths:**

The paper operationalizes the vague concept of "novelty" into a measurable quantity, moving beyond just "memorization" to include output quality. This is a critical step for evaluating LLMs on creative and scientific tasks. I know it is a limited version of novelty and many might complain about such limited definitions but this one is intuitive.

The study is thorough, investigating multiple factors including model scaling, architecture improvements (OLMo vs OLMo-2), post-training, and various inference-time methods. This provides a holistic view of what truly drives novelty.

By focusing on open-data models with indexed corpora, the study ensures its originality claims are grounded and verifiable, which is a major strength over analyses that use a generic web corpus as a proxy.

that fundamental model improvements (scale, training) push the novelty frontier while inference tweaks mostly move along it - are both well-supported and intuitive.

**Weaknesses:**

As with all work in this area, the quality scores are a proxy. A ~0.5 Spearman correlation is moderate at best and could influence the results. The harmonic mean is sensitive to low scores, so noisy quality judgments could be impactful.

The analysis is restricted to open models which, while necessary for the methodology, are not at the state-of-the-art. It remains an open question whether these trends hold for top-tier proprietary models, or even for more modest openly downloadable but not open source models

The work relies on n-gram overlap, which is a good starting point but doesn't capture semantic or stylistic novelty. The authors acknowledge this as a possible extension.

The section on sampling temperature feels... naive. The authors show the temp/novelty curve, which is fine, but they completely ignore the vast literature on modern sampling techniques. Where is the discussion of recent methods like Min-P, Top N-sigma, Top-H, or Mirostat, etc sampling? These are becoming standard practice and have complex interactions with temperature (namely they allow very high temperatures in some cases up to infinity). Treating temperature in a vacuum is a glaring omission in an otherwise thorough paper.

Worse, the analysis misses the implementation details that actually matter in production. The order in which you apply samplers (e.g., top-p, then temp) varies across frameworks like vLLM, sglang, and the default huggingface generate method. This order has a massive impact on the final distribution. Treating temperature as a pure thermodynamic concept without acknowledging the grubby reality of its implementation is an oversimplification. For a paper about generation, this lack of systems-level awareness is a miss. I will give them a pass here ONLY because literally the entire field also makes this same mistake and it's unfair to penalize them hard for it here - but it could invalidate the entire sections results...

:et's be real, the models they evaluate kind of suck. OLMo-2 and Pythia are important for the open community, and I get that you have to use them to get access to the training data for a true originality check. But it's still sad. Seeing these trends on a model that can actually perform complex creative tasks would be far more compelling. That said, the authors correctly identify this as a necessary evil, and their framework is the obvious path forward for when we have better tools to analyze SOTA models.

**Questions:**

Why was the harmonic mean chosen over other aggregations, like the geometric mean, or simply analyzing the Pareto frontier of quality vs. originality? The harmonic mean heavily penalizes low scores in either dimension right? Was this behavior explicitly desired?

Your temperature experiments show a clear trade-off. How do you think other modern sampling methods alongside higher temperature (e.g., Min-p sampling, Mirostat, etc) would affect the novelty frontier? Would they offer a better trade-off than temperature alone?

 If you were given oracle access to the training data of a frontier model like GPT-4 or Claude 3.5, which of your findings do you believe would change the most, and why?

---

> ### Author Response · Authors · 2025-11-21
> **Response to Reviewer ePsk**
>
> We appreciate your detailed review and note the following changes we made to our draft accordingly:
> - **(Weakness 1) On correlation with human judgments of quality:** As noted by the reviewer, we provide evidence that our quality scores are reliable through a user study on 3x100 examples scored by 11 different annotators. This is in line with other studies on similar tasks [1,2,3]. While not as high as more objective tasks, we find that frontier models are increasingly better at modeling user judgments (o1 < o3, Claude 3.7 < Claude 4), so we expect that this axis is more aligned with contemporary LLM progress.
> - **(Weakness 2, 6) On using open models:** We acknowledge that our current experiments are with open-data models. We detail our vision for extensions to frontier models either by working with model providers (Appendix C) or by relaxing the definition of originality (Appendix C, G.1).
> - **(Weakness 3) On semantic measures of originality:** We acknowledge that semantic measures of originality would work well, but were not aware of any methods that could scale to the size of contemporary model training datasets.  We detail this as a limitation in Appendix B and provide experiments using perplexity for originality in Appendix G.1 that show that our conclusions generalize to this measure as well. We are open to expanding our analysis further if there is a specific measure you could point us to.
> - **(Weakness 4, Question 2) On sampling techniques:** We had to limit the exploration of sampling techniques due to the time-intensive nature of our experiments. However, we acknowledge the importance of the same, and hence we add the following explicit note in our limitations section: “Finally, we restrict ourselves to simple temperature scaling due to the time-intensive nature of our experiments. An important direction of future work is more detailed analysis of variance under different sampling strategies and system-level differences in model deployment.” - L.812-814
> - **(Question 1) On the choice of harmonic mean:** The choice of aggregation measure among harmonic, arithmetic and geometric mean was essentially arbitrary, with preference given to the harmonic mean as we intended to penalize generations where either one value (originality, or quality) was low. As to why a single measure is needed vs analyzing the frontier, this was chosen mainly to align with conventions in the field where a single benchmark number is often reported.
> - **(Question 3) What might change when extending to frontier models?** This is a super interesting question. The one we would be most interested in exploring is whether findings related to model scale change when the regime is in super large models >72B. If it is indeed true that the plateauing effect we observe holds even at these scales, this could be a very impactful finding about generalization from LLMs.
>
> [1] Li, Ruizhe, et al. "Automated Creativity Evaluation for Large Language Models: A Reference-Based Approach." arXiv preprint arXiv:2504.15784 (2025).
>
> [2] Sawicki, Piotr, Marek Grze's, Daniel Brown and Fabr'icio G'oes. “Can Large Language Models Outperform Non-Experts in Poetry Evaluation? A Comparative Study Using the Consensual Assessment Technique.” ArXiv abs/2502.19064 (2025)
>
> [3] Chiang, Cheng-Han, and Hung-Yi Lee. "Can Large Language Models Be an Alternative to Human Evaluations?." Proceedings of the 61st Annual Meeting of the Association for Computational Linguistics (Volume 1: Long Papers). 2023.

---

> > ### Comment · Reviewer_ePsk · 2025-11-26
> > **Thank you for clarification.**
> >
> > Thank you for clarification on points and resolving the questions that I have. My score is already quite high, and I really like this paper, so I do want it to be accepted.

---

### Official Review · Reviewer_bdHB · 2025-10-29

**Soundness:** 3
**Presentation:** 4
**Contribution:** 2
**Rating:** 4
**Confidence:** 1

**Summary:**

This articles proposed to use a harmonic mean as a measurement of novelty of LLM and analyzed widely used methods incuding scaling, posting training etc. to observe improvement in orignnality of the generated contexts

**Strengths:**

Creativity of LLM is relatively less explored area in LLMs. Even though creativity = a balance of quality and novelty, the authors provide a more detailed quantiative framework for measurrement. And provide evaluation using a series of different experiments , including Story completion., Poetry writing. and Creative tool use. on  OLMo, OLMo-2, and Pythia at different scale. The authors calculate originality as the fraction of n-grams that do not appear in model training data and se LLM-as-a-judge to approximate the measure of output quality from human annotators.

Importantly, the author did a good job to invetigate what factors affect the novelty of LLMs, includeing sizes, quality of base model, post-training , temperature, in-context learning and prompt engineering.

**Weaknesses:**

3 weakest point:

1) definition of originality is too limited. Should at least try 1-2 other defnition and check results. the novelty score, by LLM, alone is not enough other method is needed

2) I understand why the authors start by using 3 open-data LLM. However, we also need to check other SOTA model at least the open-source ones such as LLama, deepseek, Qwen etc.

3) The contribution is only at intermediate level because there has been many work talking about creativity = balance of novelty and quality

**Questions:**

The U-shape effect of temperature is interesting , any theoretical explanation ？

novelty of reasoning model vs. non-reasoning model?

scalability of creativity by test-time computing

---

> ### Author Response · Authors · 2025-11-21
> **Response to Reviewer bdHB (1/1)**
>
> We appreciate your feedback and recognition of the strengths of our work. Here we discuss how we incorporated the same in our draft and respond to your concerns:
> - **(Weakness 1) On different measures of originality:** We define novel text as the combination of original and high-quality text. We selected n-gram originality as it is an exact measure connecting the generations to model training data. To address your concerns about different measures of originality, we provide experiments in Appendix G.1 using perplexity instead of unseen n-gram fraction. From Fig. 7, we find that our findings are robust and generalize to this new measure as well, that (a) post-training leads to higher novelty, and (b) at the same scale, a higher-quality language model also generates more novel text driven by originality. We believe that this demonstrates how our proposed framework can be adapted to more definitions of originality. However, we note two challenges with perplexity: (1) it is unbounded and hence must be normalized to the reference text from the dataset, and (2) perplexity might be high not for a variety of reasons, including data domain coverage in pre-training and incoherence, so it is not immediately clear that high perplexity text is always original potentially limiting future reuse. We include this in Appendix G.1 as well to caveat the usage.
> - **(Weakness 2) On extensions to frontier models:** We acknowledge that our evaluation is performed on open-data models. In Appendix C, we describe how this could be adapted to frontier models. We envision that model providers could include this analysis as part of model cards accompanying releases. We note that reporting aggregated novelty scores still preserves their competitive advantage (training data) while helping facilitate robust benchmarking on true generalization as models continue to improve. Noting the challenge of access to frontier labs, we could also extend our analysis by relaxing the definition of originality to a proxy measure like perplexity (described in Appendix G.1) or a representative corpus of the internet (as used in [1]). These methods have limitations, described in Appendix G.1 and L.374, Footnote 10, respectively.
> - **(Weakness 3) On the connection to related work connecting novelty and output quality:** We believe that the key difference that makes our work stand out among existing literature is the explicit connection to pre-training and post-training data (as noted by R-XcZG). By using the model training data to analyze behavior, we believe that we are able to draw strong conclusions and provide reliable benchmarking. We have included a discussion about literature that analyses the trade-offs between output quality and aspects like diversity in Related Work. We’re happy to engage in discussion and cite/compare our findings to any specific works that you think should be included.
> - **​​(Question 1) On U-shaped effect from temperature:** We believe that the underlying behavior driving this phenomenon is the fact that increasing sampling temperature leads to sampling lower likelihood sequences. Considering that models are trained on large corpora of natural language text, it is reasonable to assume that lower likelihood generations for the same prompt are of lower quality. We show that this also results in more original sentences. But these two effects offset each other, meaning that there is an optimum temperature value for each task and model that leads to the highest novelty, but this is one that has to be empirically tuned and not a sustainable method for obtaining novel generations.
> - **(Question 2) On reasoning models and test time compute:** We think that expanding our analysis to reasoning models and varying test-time compute are indeed important areas to explore. We suspect that the results might not be too materially different from OLMo-2-Instruct, which was tuned with RLVR. In Section 4.2/Figure 4, we show that RL/preference tuning recovers some of the loss in originality suffered during SFT, which leads to more novel output. We are excited to expand our analysis as more open-data reasoning models are released - we note that OLMo-3-Think was released just yesterday, the first open-data reasoning model. Our hope is that the training data is indexed on tools like Infinigram, and we can include this in the camera-ready version of our draft.
>
> [1] Lu, Ximing, et al. "AI as Humanity's Salieri: Quantifying Linguistic Creativity of Language Models via Systematic Attribution of Machine Text against Web Text." arXiv preprint arXiv:2410.04265 (2024).
>
> **Please let us know if you have any other questions that we can address. If not, we sincerely hope you can consider increasing your rating to 8 (Accept) or 6 (Borderline Accept), as we believe that our contribution is significant to the open-science LLM community and LLMs X creativity communities.**

---

### Official Review · Reviewer_XcZG · 2025-10-31

**Soundness:** 3
**Presentation:** 4
**Contribution:** 2
**Rating:** 6
**Confidence:** 4

**Summary:**

The paper demonstrates an n-gram based metric for novelty of LLM generations in a domain, where the metric is computed using n-gram statistics on the training set. The metric is not semantically aware but does combine quality and diversity in a reasonable and novel way. Evaluations are on open-source model suites with open training data (mostly Olmo 1 & 2). Interesting findings are that post-training helps novelty, prompt engineering (as far as tested) does not noticably affect it, and model scale improves things.

**Strengths:**

- Good quality experiments validating the core metric, and the paper is persuasive that post-training helps novelty significantly, which I might not have expected a priori.
- The conclusion about prompt engineering techniques yielding limited diversity is valuable, although it could be made more robust by investigating a greater variety of techniques (for example, Bradley et al., (2023) looked at poetry generation and used an evolutionary algorithm-based elicitation technique for increased diversity).

**Weaknesses:**

- The core metric the paper develops is not semantically aware (as the authors note). As a result, I am uncertain how I would usefully apply this in a practical context (e.g., scientific idea generation conditioned on the literature). Perhaps I would use it as part of a basket of other diversity metrics? The authors could clarify.
- I would have liked to see more of a focus on post-trained models, since a common debate regarding the diversity of LLM generations is whether RL (DPO, PPO, RLVR) will tend to collapse the diversity of outputs. The paper's finding that post-training helps on Olmo is good, but to make it more robust it would have been great to evaluate e.g. DeepSeek models, or another similar provider? Some of these models are significantly larger than 32B so would give much more robust conclusions—as it is, we cannot justifiably have much confidence in these conclusions generalizing well to the frontier, because there is more than an OOM of parameters between the biggest model here and the frontier.
- Comparing to training data is appropriate given this paper's goals, but for maximum usefulness it would have been good to spend more time in the main paper operationalizing how one would use this metric without reference to training data (perhaps instead with reference to a precomputed corpus/presampled corpus). After all, one of the main uses of a diversity metric in general would be to evaluate e.g. GPT-5 in a domain of choice to see how diverse its outputs are relative to either some human baseline (where that makes sense) or to other models.

**Questions:**

- Please cite Shypula et al. 2025 (https://arxiv.org/abs/2504.12522) and other similar prior work proposing diversity/quality metrics. The paper could do with better contextualizing the novelty of its contributions. For example: it is one of few works which draw on the training data statistics; it is one of few works which unify quality and diversity directly.

---

> ### Author Response · Authors · 2025-11-21
> **Response to Reviewer XcZG (1/3)**
>
> We appreciate your detailed feedback. We respond to your concerns below:
> - **(Weakness 1) On instantiating our framework on a (practical) task like scientific discovery:** Our goal was to provide a general framework for measuring the frontier of originality and output quality, instantiated in creative tasks. We chose this set of tasks as it is commonly reported that one of the most common use cases of LLMs is writing assistance [3]. Our tasks represent the documented use cases of professional writers using LLMs [4] - brainstorming alternative completions to stories (Tinystories), generating text that follows semantic and stylistic constraints (CoPoet), and reasoning about physical situations for creative plot points (MacGyver). More broadly, originality is intended to measure the distance of the generation from the training data, while quality is intended to be a task-specific measure of how well the generation satisfies the task. A natural way to adapt our framework for scientific discovery would be to create a quality rubric that incorporates a measure of the rigour of experimental evidence (to support claims) and the contribution compared to existing literature. This would not be too dissimilar to a conference reviewing rubric. This could be instantiated with a search-augmented research agent that has access to the Semantic Scholar API to evaluate the contribution, and a powerful LLM to verify the evidence presented for the claims. We include this in our revision in Discussion, L.520-522.

---

> ### Author Response · Authors · 2025-11-21
> **Response to Reviewer XcZG (2/3)**
>
> - **(Weakness 2) On the effect of different post-training methods:** We expand our analysis to include results from the intermediate checkpoint of OLMo-7B and OLMo-2-7B after the SFT stage (prior to DPO in OLMo and RLVR in OLMo-2). We find that SFT increases output quality from the base models, but at a significant cost of output originality. The preference tuning step (RL or DPO) is able to recover this loss, leading to the higher novelty scores reported in our paper. This finding expands on the results from prior work [1] that observed a similar ‘memorization-style’ improvement in performance on arithmetic reasoning and further shows the importance of studying originality and output quality in tandem.   We think these results are important, so they have been incorporated into the draft in Section 3.2 and Figure 4. Our work provides a recipe that can be replicated by model developers to analyze the trade-offs of post-training.
>
> Here is the comparison for TinyStories:
> | | Output Quality | Unique Fraction |       |       | Novelty |        |        | Novelty - Top 10  |        |        |
> |-------------------:|:--------------:|:---------------:|:-----:|:-----:|:-------:|:------:|:------:|:-----------------:|:------:|:------:|
> |                    | Output Quality |      n = 4      | n = 5 | n = 6 |  n = 4  |  n = 5 |  n = 6 |       n = 4       |  n = 5 |  n = 6 |
> |    Dataset - Dolma |      0.876     |      0.126      | 0.359 | 0.641 |  0.214  |  0.503 |  0.751 |       0.364       |  0.639 |  0.851 |
> |          OLMo-2 7B |      0.758     |      0.511      | 0.758 | 0.886 |  0.366  |  0.225 |  0.034 |        0.44       |  0.293 |  0.132 |
> |      OLMo-2-7B-SFT |      0.855     |       0.26      | 0.533 | 0.766 |   0.17  |  0.139 |  0.046 |       0.262       |  0.203 |  0.101 |
> | OLMo-2 7B Instruct |      0.936     |       0.59      | 0.837 | 0.954 |  0.503  |  0.378 |  0.191 |       0.492       |  0.325 |  0.146 |
>
>
> Similarly for CoPoet:
> | | Output Quality | Unique Fraction |       |       | Novelty |        |        | Novelty - Top 10  |        |        |
> |-------------------:|:--------------:|:---------------:|:-----:|:-----:|:-------:|:------:|:------:|:-----------------:|:------:|:------:|
> |                    | Output Quality |      n = 4      | n = 5 | n = 6 |  n = 4  |  n = 5 |  n = 6 |       n = 4       |  n = 5 |  n = 6 |
> |    Dataset - Dolma |      0.626     |      0.188      | 0.358 | 0.462 |  0.228  |  0.363 |  0.439 |       0.727       |  0.888 |  0.988 |
> |          OLMo-2 7B |      0.483     |      0.549      | 0.772 |  0.87 |  0.214  |  0.18  |  0.128 |       0.062       | -0.033 | -0.105 |
> |      OLMo-2-7B-SFT |      0.626     |      0.449      | 0.737 | 0.872 |  0.221  |  0.26  |  0.244 |       0.122       |  0.023 | -0.064 |
> | OLMo-2 7B Instruct |       0.7      |      0.834      |  0.93 | 0.926 |  0.511  |  0.409 |  0.319 |       0.208       |  0.077 | -0.015 |
>
> And for MacGyver:
> | | Output Quality | Unique Fraction |       |       | Novelty |        |        | Novelty - Top 10  |        |        |
> |-------------------:|:--------------:|:---------------:|:-----:|:-----:|:-------:|:------:|:------:|:-----------------:|:------:|:------:|
> |                    | Output Quality |      n = 4      | n = 5 | n = 6 |  n = 4  |  n = 5 |  n = 6 |       n = 4       |  n = 5 |  n = 6 |
> |    Dataset - Dolma |      0.908     |      0.359      | 0.601 | 0.803 |  0.505  |  0.728 |  0.856 |       0.629       |  0.841 |  0.966 |
> |          OLMo-2 7B |      0.519     |      0.609      |  0.85 | 0.955 |  0.001  | -0.141 |  -0.24 |       0.191       |  0.102 |  0.02  |
> |      OLMo-2-7B-SFT |      0.681     |      0.367      | 0.625 | 0.825 |  -0.064 | -0.122 | -0.157 |       0.049       |  0.033 | -0.003 |
> | OLMo-2 7B Instruct |      0.892     |      0.677      | 0.883 | 0.969 |  0.247  |  0.142 |  0.055 |       0.258       |  0.14  |  0.034 |

---

> ### Author Response · Authors · 2025-11-21
> **Response to Reviewer XcZG (3/3)**
>
> - **(Weakness 3) On extensions for when we might not have access to training data:**  We acknowledge that our evaluation is performed on open-data models. In Appendix C, we describe how this could be adapted to frontier models. We envision that model providers could include this analysis as part of model cards accompanying releases. We  note that reporting aggregated novelty scores still preserves their competitive advantage (training data) while helping facilitate robust benchmarking on true generalization as models continue to improve. Noting the challenge of access to frontier labs, we could also extend our analysis by relaxing the definition of originality to a proxy measure like perplexity (described in Appendix G.1) or a representative corpus of the internet (as used in [2]). From Fig. 7, we find that our findings are robust and generalize to this new measure as well, that (a) post-training leads to higher novelty, and (b) at the same scale, a higher-quality language model also generates more novel text driven by originality. These methods have limitations, described in Appendix G.1 and L.374, Footnote 10, respectively.
> - **(Question 1)** We thank the reviewer for the pointer and provide explicit details on the connection to works measuring output diversity and quality. We have added this reference, as well as a few more diversity-quality papers, in L.508-512.
>
> [1] Chu, Tianzhe, et al. "Sft memorizes, rl generalizes: A comparative study of foundation model post-training." arXiv preprint arXiv:2501.17161 (2025).
>
> [2] Lu, Ximing, et al. "AI as Humanity's Salieri: Quantifying Linguistic Creativity of Language Models via Systematic Attribution of Machine Text against Web Text." arXiv preprint arXiv:2410.04265 (2024).
>
> [3] Zhao, Wenting, et al. "Wildchat: 1m chatgpt interaction logs in the wild." arXiv preprint arXiv:2405.01470 (2024).
>
> [4] Chakrabarty, Tuhin, et al. "Creativity support in the age of large language models: An empirical study involving professional writers." Proceedings of the 16th Conference on Creativity & Cognition. 2024.
>
> **We hope that you are open to reconsidering your score to 8 (Accept), given the updated draft incorporating your feedback.**

---

> > ### Comment · Reviewer_XcZG · 2025-11-25
> >
> > Thank you for your contributions and updates! This is useful, and especially the additional comparisons and pointers to literature. I have updated my score to an 8 to reflect improvements.

---

### Author Response · Authors · 2025-12-01
**Summarizing the discussion and updated results**

First, we'd like to thank the AC and organising committee for stepping in with a solution in a short time span, given recent events. We also thank the reviewers for their feedback and for engaging with the responses we provided during the rebuttal period. Here, we summarise how we address/addressed the feedback of each reviewer for the benefit of the AC:
- In response to comments from Reviewer XcZg, we added experiments comparing the different stages of post-training. We find that SFT increases output quality from the base models, but at a significant cost of output originality. The preference tuning step (RL or DPO) is able to recover this loss, leading to the higher novelty scores reported for post-trained models. This finding expands on the results from prior work [1] that observed a similar ‘memorization-style’ improvement in performance on arithmetic reasoning and further shows the importance of studying originality and output quality in tandem. We think these results are important, so they have been incorporated into the draft in Section 3.2 and Figure 4. **The reviewer agreed that our new results (and other clarifications) resolved their questions and kindly updated their score to 8 (from 6), which we hope will be considered during the decision-making for our paper.**
- To address a concern from Reviewer bdHB about reporting results on alternative measures of originality, we confirmed that **we selected n-gram originality as it is an exact measure connecting the generations to model training data** (noted separately by Reviewer XcZg). To assuage their concern that our findings might be limited to unseen n-gram fraction, **we also provided experiments in Appendix G.1 using perplexity to score originality**. From Fig. 7, we find that **our findings are robust and generalise to this new measure as well**, that (a) post-training leads to higher novelty, and (b) at the same scale, a higher-quality language model also generates more novel text driven by originality. We also discussed some challenges with using perplexity as a measure so we recommend the n-gram originality measure. We did not get a chance to hear back from Reviewer bdHB, but we believe that we have assuaged many of their concerns.
- The main concern of Reviewer KYVM was about the usefulness of our metric. Our rebuttal was that a measure of novelty that jointly studies the quality and originality of output based on the training data is necessary for evaluating the true generalisation of foundation models/LLMs. Such a measure is particularly valuable in domains like creativity and scientific discovery, where it is desirable for the model to extrapolate from the training data to new insights. We also note an interesting case study [2] where **GPT-5 was believed to have helped generate a novel proof that was later found to be copied from an earlier arxiv paper---we think that this provides direct motivation for evaluations that connect model output to training data such as ours.**
- We thank Reviewer ePsk for their positive review score of 8 as well as their concrete feedback. We included a discussion about system-level variance in sampling techniques in the paper based on their comments.

[1] Chu, Tianzhe, et al. "Sft memorizes, rl generalizes: A comparative study of foundation model post-training." arXiv preprint arXiv:2501.17161 (2025).

[2] "Early science acceleration experiments with GPT-5", https://cdn.openai.com/pdf/4a25f921-e4e0-479a-9b38-5367b47e8fd0/early-science-acceleration-experiments-with-gpt-5.pdf

---

### Meta-Review · Area_Chair_g2wH · 2026-01-07

**Summary:**

This paper proposes a concrete, data-grounded framework for measuring the novelty of LLM generations by explicitly combining originality and quality. Originality is defined via n-gram overlap with the model’s training data, while quality is measured using an LLM-as-a-judge; the two are aggregated using a harmonic mean. The authors conduct an extensive empirical study on open-data models across multiple creative tasks, examining the effects of model scale, base model quality, post-training, and inference-time techniques.

Reviewers agree that the paper is well written, technically sound, and empirically thorough. The strongest aspect of the work is the systematic operationalization of a previously vague concept—novelty—into a measurable quantity that explicitly balances originality and quality, grounded in verifiable training data. Multiple reviewers found the experimental analysis persuasive, especially the consistent finding that model improvements and post-training (rather than prompt engineering or sampling tricks) are the primary drivers of higher novelty.

At the same time, reviewers also raised concerns about the scope and generality of the proposed metric. The reliance on n-gram overlap is acknowledged as limited and not semantically aware, and the evaluation is necessarily restricted to open-data models that are far from the frontier. Several reviewers questioned the practical usefulness of the metric outside of this controlled setting and whether the harmonic mean is the most principled way to aggregate originality and quality.

The authors provided a strong and detailed rebuttal, addressing most major concerns. They expanded analyses to intermediate post-training stages, demonstrated robustness to alternative originality proxies, clarified the intended use cases of the framework, improved contextualization within prior work, and explicitly discussed limitations regarding frontier models and sampling strategies. These changes convinced at least one initially borderline reviewer to raise their score substantially, and strengthened the paper overall.

Overall, this submission makes a meaningful contribution to the study of creativity and novelty in LLMs by offering a concrete, reproducible, and data-centric evaluation framework, even if the metric itself is necessarily imperfect. While concerns about semantic novelty and generalization remain valid, they are clearly acknowledged and do not undermine the paper’s core empirical insights. Given the solid experimental evidence, clear presentation, and improved revision, I recommend acceptance.

**Reviewer Concerns:**

see above

**Reviewer Scores:**

see above

---

### Decision · Program_Chairs · 2026-01-26

Accept (Poster)